# Energy Matching: Unifying Flow Matching and Energy-Based Models for Generative Modeling

**Michal Balcerak**
University of Zurich
michal.balcerak@uzh.ch

**Tamaz Amiranashvili**
University of Zurich
Technical University of Munich

**Antonio Terpin**
ETH Zurich

**Suprosanna Shit**
University of Zurich

**Lea Bogensperger**
University of Zurich

**Sebastian Kaltenbach**
Harvard University

**Petros Koumoutsakos**
Harvard University

**Bjoern Menze**
University of Zurich

## Abstract

Current state-of-the-art generative models map noise to data distributions by matching flows or scores. A key limitation of these models is their inability to readily integrate available partial observations and additional priors. In contrast, energy-based models (EBMs) address this by incorporating corresponding scalar energy terms. Here, we propose *Energy Matching*, a framework that endows flow-based approaches with the flexibility of EBMs. Far from the data manifold, samples move from noise to data along irrotational, optimal transport paths. As they approach the data manifold, an entropic energy term guides the system into a Boltzmann equilibrium distribution, explicitly capturing the underlying likelihood structure of the data. We parameterize these dynamics with a single time-independent scalar field, which serves as both a powerful generator and a flexible prior for effective regularization of inverse problems. The present method substantially outperforms existing EBMs on CIFAR-10 and ImageNet generation in terms of fidelity, while retaining simulation-free training of transport-based approaches away from the data manifold. Furthermore, we leverage the flexibility of the method to introduce an interaction energy that supports the exploration of diverse modes, which we demonstrate in a controlled protein generation setting. This approach learns a scalar potential energy, without time conditioning, auxiliary generators, or additional networks, marking a significant departure from recent EBM methods. We believe this simplified yet rigorous formulation significantly advances EBMs capabilities and paves the way for their wider adoption in generative modeling in diverse domains.

## 1 Introduction

Generative models learn to map from a simple, easy-to-sample distribution, such as a Gaussian, to a desired data distribution. They do so by approximating the optimal transport (OT) map—such as in flow matching [Lipman et al., 2023, Liu et al., 2023, Albergo and Vanden-Eijnden, 2023]—or through iterative noising and denoising schemes, such as in diffusion models [Ho et al., 2020, Song et al., 2021]. In addition to being highly effective in sample generation, diffusion- and flow-based models have also been used as priors to regularize poorly posed inverse problems [Chung et al., 2023, Mardani et al., 2024, Ben-Hamu et al., 2024]. However, these models do not explicitly capture the unconditional data score and instead model the score of smoothed manifolds at different noise levels. The measurement likelihood, on the other hand, is not tractable on these noised manifolds. As a

| Action Matching | OT-Flow Matching | Energy-Based Model | Energy Matching (Ours) |

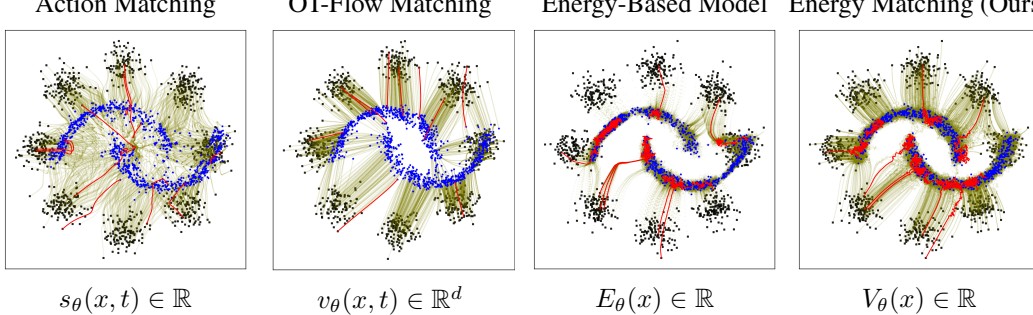

$$s_\theta(x,t) \in \mathbb{R} \qquad v_\theta(x,t) \in \mathbb{R}^d \qquad E_\theta(x) \in \mathbb{R} \qquad V_\theta(x) \in \mathbb{R}$$

Figure 1: Trajectories (green lines) of samples traveling from a noise distribution (black dots; here, a Gaussian mixture model) to a data distribution (blue dots; here, two moons as in [Tong et al., 2023]) under four different methods: Action Matching [Neklyudov et al., 2023], Flow Matching (OT-CFM) [Tong et al., 2023], EBMs trained via contrastive divergence [Hinton, 2002], and our proposed Energy Matching. We highlight several individual trajectories in red to illustrate their distinct behaviors. Both Action Matching and Flow Matching learn time-dependent transports and are not trained for traversing the data manifold. Conversely, EBMs and Energy Matching are driven by time-independent fields that can be iterated indefinitely, allowing trajectories to navigate across modes. While samples from EBMs often require additional steps to equilibrate (see, e.g., the visible mode collapses that slow down sampling from the data manifold), Energy Matching directs samples toward the data distribution in "straight" paths, without hindering the exploration of the data manifold.

result, existing approaches repeatedly shuttle between noised and data distributions, leading to crude approximations of complex, intractable terms Daras et al. [2024]. For example, DPS [Chung et al., 2023] approximates an intractable integral using a single sample. More recently, D-Flow [Ben-Hamu et al., 2024] optimizes initial noise by differentiating through the simulated trajectory. To the best of our knowledge, these models lack a direct way to navigate the data manifold in search of the optimal solution without repeatedly transitioning between noised and data distributions.

EBMs [Hopfield, 1982, Hinton, 2002, LeCun et al., 2006] provide an alternative approach for approximating the data distribution by learning a scalar-valued function $E(x)$ that specifies an *unnormalized* density $p(x) \propto \exp(-E(x))$. Rather than explicitly mapping noise samples onto the data manifold, EBMs assign low energies to regions of high data concentration and high energy elsewhere. This defines a Boltzmann distribution from which one can sample, for example, via *Langevin sampling*. In doing so, EBMs explicitly retain the likelihood information in $E(x)$. This likelihood information can then be used in conditional generation (e.g., to solve inverse problems), possibly together with additional priors simply by adding their energy terms [Du and Mordatch, 2019]. Moreover, direct examination of local curvature on the data manifold—allows the computation of local intrinsic dimension (LID) (an important proxy for data complexity)—whereas diffusion models can only approximate such curvature in the proximity of noise samples.

Despite the theoretical elegance of using a *single*, time-independent scalar energy, practical EBMs have historically suffered from poor generation quality, falling short of the performance of diffusion or flow matching models. Traditional methods [Song and Kingma, 2021] for training EBMs, such as contrastive divergence via Markov chain Monte Carlo (MCMC) or local score-based approaches [Song and Ermon, 2019], often fail to adequately explore the energy landscape in high-dimensional spaces, leading to instabilities and mode collapse. Consequently, many methods resort to time-conditioned ensembles [Gao et al., 2021], hierarchical latent ensembles [Cui and Han, 2024], or combine EBMs with separate generator networks trained in cooperation [Guo et al., 2023, Zhang et al., 2024, Yoon et al., 2024], thereby requiring significantly higher parameter counts and training complexity.

**Contributions.** In this work, we propose *Energy Matching*, a two-regime training strategy that combines the strengths of EBMs and flow matching; see Figure 1.

When samples lie far from the data manifold, they are efficiently transported toward the data. Once near the data manifold, the flow transitions into Langevin steps governed by an internal energy component, enabling precise exploration of the Boltzmann-like density well around the data distribution. This straightforward approach produces a *time-independent* scalar energy field whose gradient both accelerates sampling and shapes the final density well—via a contrastive objective that directly learns the score at the data manifold—yet remains efficient and stable to train. Empirically, our method significantly outperforms existing EBMs on both CIFAR-10 and ImageNet generation in terms of fidelity, and compares favorably to flow-matching and diffusion models—without auxiliary generators or time-dependent EBM ensembles.

Our proposed process complements the advantages of flow matching with an explicit likelihood modeling, enabling traversal of the data manifold without repeatedly shuffling between noise and data distributions. This simplifies both inverse problem solving and controlled generations under a prior. In addition, to encourage diverse exploration of the data distribution, we showcase how repulsive interaction energies can be easily and effectively incorporated, with an application to conditional protein generation. Finally, we also showcase how analyzing the learned energy reveals insight on the LID of the data with fewer approximations than diffusion models.[a]

---

[a] Code repository: `https://github.com/m1balcerak/EnergyMatching`

## 2 Energy matching

In this section, we show how a scalar potential $V(x)$ can simultaneously provide an optimal-transport-like flow from noise to data while also yielding a Boltzmann distribution that explicitly captures the unnormalized log-likelihood of the data.

**The Jordan–Kinderlehrer–Otto (JKO) scheme.** The starting point of our approach is the JKO scheme [Jordan et al., 1998], which is the basis of the success of numerous recent generative models [Xu et al., 2023, Terpin et al., 2024, Choi et al., 2024]. The JKO scheme describes the discrete-time evolution of a probability distribution $\rho_t$ along energy-minimizing trajectories in the Wasserstein space,

$$\rho_{t+\Delta t} = \arg\min_{\rho} \underbrace{\frac{W_2^2(\rho, \rho_t)}{2\Delta t}}_{\text{Transport Cost}} + \underbrace{\int V_\theta(x)\mathrm{d}\rho(x)}_{\text{Potential Energy}} + \underbrace{\varepsilon(t)\int \rho(x)\log\rho(x)\mathrm{d}x}_{\text{Internal Energy (-Entropy)}}. \quad (1)$$

Here, $\theta$ denotes the learnable parameters of the scalar potential $V_\theta(x)$, and $\varepsilon(t)$ is a temperature-like parameter tuning the entropic term. The transport cost is given by the Wasserstein distance,

$$W_2^2(\rho, \rho_t) = \min_{\gamma \in \Gamma(\rho, \rho_t)} \int_{\mathbb{R}^d \times \mathbb{R}^d} \|x - y\|^2 \mathrm{d}\gamma(x, y), \quad (2)$$

where $\Gamma(\rho, \rho_t)$ is the set of couplings between $\rho$ and $\rho_t$, i.e., the set of probability distributions on $\mathbb{R}^d \times \mathbb{R}^d$ with marginals $\rho$ and $\rho_t$. Here, $d$ is the dimensionality of the data. Henceforth, we call OT coupling any $\gamma_t$ that yields the minimum in (2). When $\gamma_t = (\mathrm{id}, T)_{\#}\rho$, i.e., it is the *pushforward* of the map $x \mapsto (x, T(x))$ for some function $T$, we say that $T$ is an OT map from $\rho$ to $\rho_t$.

Differently from most literature, we consider $\varepsilon(t)$ to be dependent on time and study the behavior of Equation (1) as $t \to \infty$. To fix the ideas, consider, for instance, a linear scheduling:

$$\varepsilon(t) = \begin{cases} 0, & 0 \le t < \tau^*, \\ \varepsilon_{\max}\frac{t-\tau^*}{1-\tau^*}, & \tau^* \le t < 1, \\ \varepsilon_{\max}, & t \ge 1. \end{cases} \quad (3)$$

**First-order optimality conditions.** Following Terpin et al. [2024], we analyze (1) at each time $t$ via its first-order optimality conditions [Lanzetti et al., 2024, 2025]. These conditions characterize the properties of the desired solution and thus represent the optimization goal:

$$\frac{1}{\Delta t}(x - y) + \nabla_x V_\theta(x) + \varepsilon(t)\nabla_x \log \rho_{t+\Delta t}(x) = 0, \qquad (x, y) \in \mathrm{supp}(\gamma_t) \qquad (4)$$

where $\gamma_t$ is an OT plan between the distributions $\rho_{t+\Delta t}$ and $\rho_t$ and $\mathrm{supp}(\gamma_t)$ is the support of $\gamma_t$. That is, this condition has to hold for all pairs of points in the support of $\rho_{t+\Delta t}$ and $\rho_t$ that are coupled by OT. Intuitively, analyzing (4) provides us with two key insights:

1. For times $t < \tau^*$, $\varepsilon(t) = 0$ and (4) becomes

$$\frac{1}{\Delta t}(x - y) + \nabla_x V_\theta(x) = 0 \qquad (x, y) \in \mathrm{supp}(\gamma_t). \qquad (5)$$

   That is, the system is in an OT, flow-like, regime.

2. Near the data manifold, which we aim at modeling with the equilibrium distribution $\rho_{\mathrm{eq}}$ of (1), $\rho_{t+\Delta t} \approx \rho_{\mathrm{eq}}$ and, thus, for $t \gg 1$, $x \approx y$ for all $(x, y) \in \mathrm{supp}(\gamma_t)$. Then, we can simplify (4) as

$$\varepsilon_{\max}\nabla_x \log \rho_{\mathrm{eq}}(x) = -\nabla_x V_\theta(x) \quad \implies \quad \rho_{\mathrm{eq}}(x) \propto \exp\left(-\frac{V_\theta(x)}{\varepsilon_{\max}}\right).$$

   Thus, the equilibrium distribution is described by an EBM, $\exp(-E(x))$, with $E(x) = \frac{V_\theta(x)}{\varepsilon_{\max}}$.

**Our approach in a nutshell.** Combining the two insights above, we propose a generative framework that combines OT and EBMs to learn a *time-independent* scalar potential $V_\theta(x)$ whose Boltzmann distribution,

$$\rho_{\mathrm{eq}}(x) \propto \exp\left(-\frac{V_\theta(x)}{\varepsilon_{\max}}\right), \qquad (6)$$

matches $\rho_{\mathrm{data}}$. To transport samples efficiently from noise $\rho_0$ to $\rho_{\mathrm{eq}} \approx \rho_{\mathrm{data}}$, we use two regimes:

- *Away from the data manifold*: $\varepsilon \approx 0$. The flow is deterministic and OT-like, allowing rapid movement across large distances in sample space.

- *Near the data manifold*: $\varepsilon \approx \varepsilon_{\max}$. Samples diffuse into a stable Boltzmann distribution, properly covering all data modes.

By combining the long-range transport capability of flows with the local density modeling flexibility of EBMs, we achieve tractable sampling and explicitly encode the unnormalized log-likelihood $-V_\theta(x)/\varepsilon_{\max}$ of the underlying data distribution; see Figure 1.

## 2.1 Training objectives

In practice, we balance the two objectives by initially training $V_\theta$ exclusively with the optimal-transport-like objective ($\varepsilon = 0$, see Section 2.1.1), ensuring a stable and consistent generation of high-quality negative samples for the contrastive phase. Subsequently, we jointly optimize both the transport-based and contrastive divergence objectives, progressively increasing the effective temperature to $\varepsilon = \varepsilon_{\max}$ as samples approach the data manifold (i.e., the equilibrium distribution); see Section 2.1.2.

### 2.1.1 Flow-like objective $\mathcal{L}_{\mathrm{OT}}$

We begin by constructing a global velocity field $-\nabla_x V_\theta(x)$ that carries noise samples $\{x_0\}$ to data samples $\{x_{\mathrm{data}}\}$ with minimal detours. For this, we consider geodesics in the Wasserstein space [Ambrosio et al., 2008]. Practically, we compute the OT coupling $\gamma^*$ between two uniform empirical probability distributions, one supported on a mini-batch of the data, and one supported on a set of noise samples with the same cardinality. These samples are drawn from an easy-to-sample distribution; in our case, a Gaussian. Since the probability distributions are uniform and empirical with the same number of samples, a transport map $T$ is guaranteed to exist [Ambrosio et al., 2008].

**Remark 2.1** (OT solver). *Depending on the method used to compute the OT coupling, an explicit OT map may or may not be obtained. Similarly, if the number of noise samples differs from the mini-batch size $B$, the resulting OT coupling generally will not correspond to a map. In this case, one can adapt the algorithm by defining a threshold $\pi_{\mathrm{th}}$ and considering all pairs $(x_{\mathrm{data}}, x_0)$ for which the coupling value satisfies $\gamma^*(x_{\mathrm{data}}, x_0) > \pi_{\mathrm{th}}$. In our experiments, we used the POT solver [Flamary et al., 2021] and did not observe benefits from using a sample size different from $B$, consistent with previous approaches [Tong et al., 2023].*

Then, for each data point $x_{\text{data}}$ we define the *interpolation* $x_t = (1 - t)T(x_{\text{data}}) + tx_{\text{data}}$, which is a point along the geodesic. The *velocity* of each $x_t$ is $x_{\text{data}} - T(x_{\text{data}})$ (i.e., the samples move from the noise to the data distribution at constant speed) and, in this regime, we would like to have $-\nabla_x V_\theta(x_t) \approx x_{\text{data}} - T(x_{\text{data}})$. For this, we define the loss:

$$\mathcal{L}_{\text{OT}} = \mathbb{E}_{t \sim U(0,\tau^*)}^{x_{\text{data}} \in \mathcal{D}} \left[ \|\nabla_x V_\theta(x_t) + x_{\text{data}} - T(x_{\text{data}})\|^2 \right].$$

This objective can be interpreted as a flow-matching formulation under the assumption that the velocity field is both time-independent and given by the gradient of a scalar potential, thereby imposing an irrotational condition. This aligns naturally with OT, which also yields an irrotational velocity field—any rotational component would add unnecessary distance to the flow and thus inflate the transport cost without benefit. Our experimental evidence adds to the recent study by [Sun et al., 2025], in which the authors observed that time-independent velocity fields can, under certain conditions, outperform time-dependent noise-conditioned fields in sample generation.

---

**Algorithm 1** Phase 1 (warm-up).

---

1:  **Initialize** model parameters $\theta$
2:  **for** iteration $n = 0, 1, \ldots$ **do**
3:      Sample mini-batch $\{x_{\text{data},b}\}_{b=1}^B \sim \mathcal{D}$                               $\triangleright$ Data samples
4:      Sample mini-batch $\{x_{0,b}\}_{b=1}^B \sim \mathcal{N}(0, I)$                  $\triangleright$ Random Gaussian samples
5:      $T \leftarrow \text{OTsolver}(\{x_{\text{data},b}\}, \{x_{0,b}\})$                     $\triangleright$ Compute OT map
6:      Sample $\{t_b\}_{b=1}^B \sim U(0, \tau^*)$                   $\triangleright$ Typically $\tau^* = 1$ for the warm-up
7:      Set interpolations $x_{t_b} \leftarrow (1 - t_b)\, T(x_{\text{data},b}) + t_b\, x_{\text{data},b}$      $\triangleright$ Interpolation along geodesics
8:      $\mathcal{L}_{\text{OT}}(\theta) \leftarrow \sum_{b=1}^B \|\nabla_x V_\theta(x_{t_b}) + x_{\text{data},b} - T(x_{\text{data},b})\|^2$      $\triangleright$ Loss function
9:      $\theta \leftarrow \theta - \alpha \nabla_\theta \mathcal{L}_{\text{OT}}(\theta)$             $\triangleright$ Gradient update with learning rate $\alpha$
10: **end for**
11: **return** $\theta$                                         $\triangleright$ Trained $\theta$

---

### 2.1.2 Contrastive objective $\mathcal{L}_{\text{CD}}$

Near the data manifold, $V_\theta(x)$ is refined so that $\rho_{\text{eq}}(x) \propto \exp\left(-V_\theta(x)/\varepsilon_{\max}\right)$ matches the data distribution. We adopt the contrastive divergence loss described in EBMs [Hinton, 2002],

$$\mathcal{L}_{\text{CD}} = \mathbb{E}_{x \sim p_{\text{data}}} \left[ \frac{V_\theta(x)}{\varepsilon_{\max}} \right] - \mathbb{E}_{\tilde{x} \sim \text{sg}(p_{\text{eq}})} \left[ \frac{V_\theta(\tilde{x})}{\varepsilon_{\max}} \right],$$

where $\tilde{x}$ are "negative" samples of the equilibrium distribution induced by $V_\theta$. We approximate these samples using an MCMC Langevin chain [Welling and Teh, 2011]. We split the initialization for negative samples: half begin at real data, and half begin at the noise distribution. This way, $V_\theta(x)$ forms well-defined basins around high-density regions while also shaping regions away from the manifold, correcting the generation. The $\text{sg}(\cdot)$ indicates a *stop-gradient* operator, which ensures gradients do not back-propagate through the sampling procedure.

### 2.1.3 Dual objective and implementation notes

To balance the deterministic flow-like regime (where $\varepsilon \approx 0$) away from the data manifold and the stochastic Boltzmann regime (where $\varepsilon \approx \varepsilon_{\max}$) near equilibrium, we adopt the linear temperature schedule described in (3). We introduce a dataset-specific hyperparameter $\lambda_{\text{CD}}$ to stabilize the contrastive objective by appropriately weighting $\mathcal{L}_{\text{CD}}$ relative to $\mathcal{L}_{\text{OT}}$. The resulting algorithm is described in detail in Algorithm 1 and Algorithm 2. Since Algorithm 2 benefits from high-quality negatives, we begin with Algorithm 1 (and, thus, with $\mathcal{L}_{\text{OT}}$ only) to ensure sufficient mixing of noise-initialized negatives.

Given the trained models, we define a sampling time $\tau_{\text{s}}$. Although convergence to the equilibrium distribution is guaranteed only as $\tau_{\text{s}} \to \infty$, we empirically observe that sample quality (measured with Fréchet inception distance (FID)) plateaus by $\tau_{\text{s}} = 3.25$ on CIFAR-10; see Section A.2. The sampling procedure, which optionally includes conditional and interaction terms, is detailed in Algorithm 3. In practice, we implement training using explicit Euler–Maruyama updates and sampling with an Euler–Heun predictor-corrector scheme, while for simplicity the algorithms illustrate only explicit updates. Additionally, the constant factor $1/\varepsilon_{\max}$ in $\mathcal{L}_{\text{CD}}$ is absorbed into $\lambda_{\text{CD}}$.

Section A.1 discusses how the landscape of $V_\theta$ evolves across these two phases. Hyperparameters for each dataset, along with intuitions to guide their selection for new datasets, are provided in Section D.

**Algorithm 2** Phase 2 (main training).

1: $\theta \leftarrow \theta_{\text{pretrained}}$            ▷ Initialize from Algorithm 1
2: **for** iteration $n = 0, 1, \dots$ **do**
3:      $\mathcal{L}_{\text{OT}} \leftarrow$ Use lines 3–8 from Algorithm 1
4:      Initialize negative samples $\{x^{(0)}_{\text{neg},b}\}^{B}_{b=1}$ from noise and/or data      ▷ Negative samples
5:      **for** $m = 0, 1, \dots, M_{\text{Langevin}} - 1$ **do**
6:          **for** $b = 1$ to $B$ **do**
7:          $\varepsilon^{(m)} \leftarrow \begin{cases} \varepsilon_{\max}, & \text{if initialized from data (Optional)} \\ \varepsilon(m\Delta t) \text{ from (3)}, & \text{otherwise} \end{cases}$
8:          Sample $\eta_b \sim \mathcal{N}(0, I)$
9:          $x^{(m+1)}_{\text{neg},b} \leftarrow x^{(m)}_{\text{neg},b} - \Delta t \nabla_x V_{\text{sg}(\theta)}(x^{(m)}_{\text{neg},b}) + \sqrt{2\Delta t \varepsilon^{(m)}}\, \eta_b$      ▷ Langevin dynamics step
10:          **end for**
11:      **end for**
12:      $\mathcal{L}_{\text{CD}} \leftarrow \frac{1}{B}\sum^{B}_{b=1}\left[V_\theta(x_{\text{data},b}) - V_\theta(x^{(M_{\text{Langevin}})}_{\text{neg},b})\right]$      ▷ Contrastive divergence loss
13:      $\mathcal{L}(\theta) \leftarrow \mathcal{L}_{\text{OT}} + \lambda_{\text{CD}}\, \mathcal{L}_{\text{CD}}$
14:      Update $\theta \leftarrow \theta - \alpha \nabla_\theta \mathcal{L}(\theta)$      ▷ Gradient descent step
15: **end for**
16: **return** $\theta$      ▷ Trained $\theta$

Table 1: FID↓ score comparison for unconditional CIFAR-10 generation (lower is better). Unless otherwise specified, we use results for solvers that most closely match our setup (325 fixed-step Euler–Heun [Butcher, 2016]). * indicates reproduced methods, while all other entries reflect the best reported results. EGC in its unconditional version has been reported in [Zhu et al., 2024]

| Learning Unnormalized Data Likelihood | | Learning Transport/Score Along Noised Trajectories | |
|---|---|---|---|
| **Ensembles: Diffusion + (one or many) EBMs** | | **Diffusion Models** | |
| Hierarchical EBM Diffusion [Cui and Han, 2024] | 8.93 | DDPM* [Ho et al., 2020] | 6.45 |
| EGC [Guo et al., 2023] | 5.36 | DDPM++ *(62M params, 1000 steps)* [Kim et al., 2021] | 3.45 |
| Cooperative DRL *(40M params)* [Zhu et al., 2024] | 4.31 | NCSN++ *(107M params, 1000 steps)* [Song et al., 2021] | 2.45 |
| Cooperative DRL-large *(145M params)* [Zhu et al., 2024] | 3.68 | | |
| **Energy-based Models** | | **Flow-based Models** | |
| ImprovedCD [Du et al., 2021] | 25.1 | Action Matching [Neklyudov et al., 2023] | 10.07 |
| CLEL-large *(32M params)* [Lee et al., 2023] | 8.61 | Flow-matching [Lipman et al., 2023] | 6.35 |
| Energy Matching *(50M params, **Ours**)* | 3.34 | OT-CFM* *(37M params)* [Tong et al., 2023] | 4.04 |

# 3 Applications

In this section, we demonstrate the effectiveness and versatility of our proposed Energy Matching approach across three applications: (i) unconditional generation (ii) inverse problems, and (iii) LID estimation. The model architecture and all the training details are reported in Section D.

## 3.1 Unconditional generation

We compare four classes of generative models: (1) Diffusion models, which deliver state-of-the-art quality but typically require many sampling steps; (2) Flow-based methods, which learn OT paths for more efficient sampling with fewer steps; (3) EBMs, which directly model the log-density as a scalar field, offering flexibility for inverse problems and constraints but sometimes at the expense of sample quality; and (4) Ensembles (Diffusion with one or many EBMs), which combine diffusion's robust sampling with elements of EBM flexibility but can become large and complex to train. Our approach, Energy Matching, offers a simple (a single time-independent scalar field) yet powerful EBM-based framework. We evaluate our approach on CIFAR-10 [Krizhevsky and Hinton, 2009] and ImageNet32x32 [Deng et al., 2009, Chrabaszcz et al., 2017] datasets, reporting FID scores in Table 1 and Table 2, respectively. Our method outperforms state-of-the-art EBMs, reducing the FID score by more than 50%.

Table 2: FID↓ score comparison for unconditional ImageNet 32x32 generation (lower is better). Unless otherwise specified, we use results for solvers that most closely match our setup (300 fixed-step Euler–Heun [Butcher, 2016]).

| Learning Unnormalized Data Likelihood | | Learning Transport/Score Along Noised Trajectories | |
|---|---|---|---|
| **Ensembles: Diffusion + (one or many) EBMs** | | **Diffusion Models** | |
| Cooperative DRL *(40M params)* [Zhu et al., 2024] | 9.35 | DDPM++ *(62M params, 1000 steps)* [Kim et al., 2021] | 8.42 |
| **Energy-based Models** | | **Flow-based Models** | |
| ImprovedCD [Du et al., 2021] | 32.48 | Flow-matching [Lipman et al., 2023] *(196M params)* | 5.02 |
| CLEL-base [Lee et al., 2023] *(7M params)* | 22.16 | | |
| CLEL-large [Lee et al., 2023] *(32M params)* | 15.47 | | |
| Energy Matching *(50M params, **Ours**)* | 6.64 | | |

## 3.2 Inverse problems

In many practical applications, we are interested in recovering some data $x$ from noisy measurements $y$ generated by an operator $A$, $y = A(x) + w$, where $w \sim \mathcal{N}(0, \sqrt{2}\,\zeta I)$. In this setting, the posterior distribution of $x$ given $y$ is

$$p(x|y) \propto \underbrace{\exp\left(-\frac{1}{\zeta^2}\|y - A(x)\|^2\right)}_{\propto\, p(y|x)} \underbrace{\exp\left(-E_\theta(x)\right)}_{\propto\, p(x)}, \qquad (7)$$

where $E_\theta(x)$ is an energy function which one can learn from the data, an EBM. Because we want to sample $x$ given a measurement $y$, this reconstruction task is often referred to as an *inverse problem*. Here, $\|y - Ax\|^2$ encodes the measurement fidelity with $\zeta$ controlling the balance between this fidelity term and the prior. We obtain the prior term $E_\theta(x) = \frac{V_\theta(x)}{\varepsilon_{\max}}$ by training $V_\theta(x)$ via Energy Matching. Samples from this posterior can be drawn by starting from a random sample $x^{(0)} \sim \mathcal{N}(0, I)$ and following a Langevin update. We detail the algorithm for generating solutions to inverse problems in Algorithm 3 (which also incorporates additional interaction energy $W(x, x')$ between generated samples). We demonstrate our model's capabilities qualitatively through a controlled inpainting task and quantitatively via a protein inverse design benchmark. Specific hyperparameters are detailed in Section D.

---

**Algorithm 3** Unconditional/conditional sampling with optional interaction energy

---

1: **for** $m = 1$ to $M$ **do**
2:      Initialize $x_m^{(0)}$ from noise and/or data     ▷ Initialize each chain
3: **end for**
4: $N \leftarrow \lfloor \tau_{\mathrm{s}} / \Delta t \rfloor$     ▷ Number of Langevin steps for sampling time $\tau_{\mathrm{s}}$
5: **for** $n = 0, 1, \ldots, N - 1$ **do**
6:      **for** $m = 1, 2, \ldots, M$ **do**     ▷ Prior + data fidelity + interaction
7:          $\varepsilon^{(n)} \leftarrow \begin{cases} \varepsilon_{\max}, & \text{if initialized from data (Optional)} \\ \varepsilon(n\Delta t) \text{ from (3)}, & \text{otherwise} \end{cases}$
8:          $U_\theta\big(x_m^{(n)}\big) \leftarrow V_\theta\big(x_m^{(n)}\big) + \varepsilon^{(n)}\big\|y - A\big(x_m^{(n)}\big)\big\|^2 / \zeta^2 + \varepsilon^{(n)} \sum_{k \neq m} W\big(x_m^{(n)}, x_k^{(n)}\big)$
9:          Sample $\eta_m^{(n)} \sim \mathcal{N}(0, I)$
10:          $x_m^{(n+1)} \leftarrow x_m^{(n)} - \Delta t\, \nabla_x U_\theta\big(x_m^{(n)}\big) + \sqrt{2\varepsilon^{(n)}\Delta t}\, \eta_m^{(n)}$     ▷ Langevin dynamics step
11:      **end for**
12: **end for**
13: **return** $\{x_m^{(N)}\}_{m=1}^M$     ▷ Final samples

---

**Controlled inpainting.** Suppose we want to recover two images from a masked image while encouraging diverse reconstructions. EBMs allow this by introducing an additional interaction energy, $W(x_1, x_2) = -\frac{\|B(x_1 - x_2)\|^2}{\sigma^2}$, where $B$ has ones in the region of interest (focusing diversity there) and zeros elsewhere, and $\sigma$ is a hyperparameter controlling the interaction's strength. Specifically, we define $p(x_1, x_2 \mid y) \propto p(x_1 \mid y)\, p(x_2 \mid y) \exp\big(-W(x_1, x_2)\big)$, which gives high probability to pairs $(x_1, x_2)$ that lie far apart in the specified region $B$. This encourages exploring the edges of the

posterior rather than just its modes, and with suitable $W$, samples shift toward rare events without needing many draws. To illustrate the interaction term's advantages for diverse reconstruction, we apply our method to a CelebA [Liu et al., 2015] $64 \times 64$ inpainting task. As shown in Figure 2, we start from a partially observed (masked) face and aim to reconstruct two distinct high-fidelity completions.

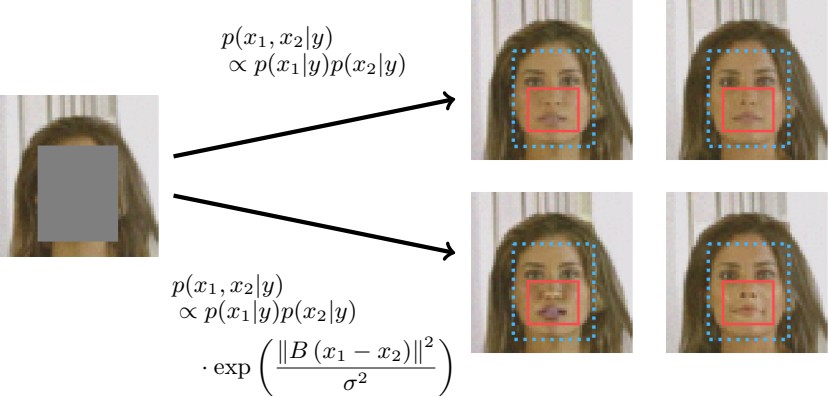

Figure 2: Controlled inpainting for diverse reconstructions. On the left is the masked face. On the right are two reconstructions: the top pair without the interaction term and the bottom pair with it. The interaction term applies in the solid red square (where $B$ has ones), and the measurement matrix $A$ is the dotted blue square (zeros inside, ones outside). By encouraging $x_1$ and $x_2$ to differ in the target region, the interaction yields a wider range of completions while preserving fidelity.

**Protein inverse design.** In Figure 3, we demonstrate our method's performance on the inverse design problem of generating Adeno-Associated Virus (AAV) capsid protein segments [Bryant et al., 2021]. Given a desired functional property (fitness)—here defined as the predicted viral packaging efficiency normalized between 0 and 1—the goal is to design novel protein sequences satisfying this target condition. Beyond achieving high fitness, practical inverse design requires generating diverse candidate sequences to ensure robustness in the downstream experimental validation [Jain et al., 2022]. We evaluate on two benchmark splits (*medium* and *hard*), which correspond to subsets of the original AAV dataset differing in baseline fitness distributions and required mutational distance from known high-performing variants [Kirjner et al., 2024]. Leveraging the latent-space representation of VLGPO [Bogensperger et al., 2025], we employ our Energy-Matching Langevin sampler with an inference-time tunable repulsion term, allowing explicit control over the diversity of the designed proteins. This enables a flexible trade-off between fitness and diversity, resulting in high fitness scores alongside substantially improved sequence diversity. See Section B for experimental details and dataset descriptions.

### 3.3 Local intrinsic dimension estimation

Real-world datasets, despite displaying a high number of variables, can often be represented by lower-dimensional manifolds—a concept referred to as the *manifold hypothesis* [Fefferman et al., 2016]. The dimension of such a manifold is called the intrinsic dimension. Estimating the LID at a given point reveals its effective degrees of freedom or *directions of variation*, offering insight into data complexity and adversarial vulnerabilities. We defer the precise definition to Section C.

**Diffusion-based approaches.** Recent work leverages pretrained *diffusion models* to estimate the LID [Kamkari et al., 2024, Stanczuk et al., 2024] by examining the learned score function. However, since these models do not learn the score at the data manifold ($t = 1$), their estimates become unreliable there. Consequently, current methods rely on approximations, for instance by evaluating the score in the proximity of the data manifold ($t = 1 - t_0$), where computations remain sufficiently reliable.

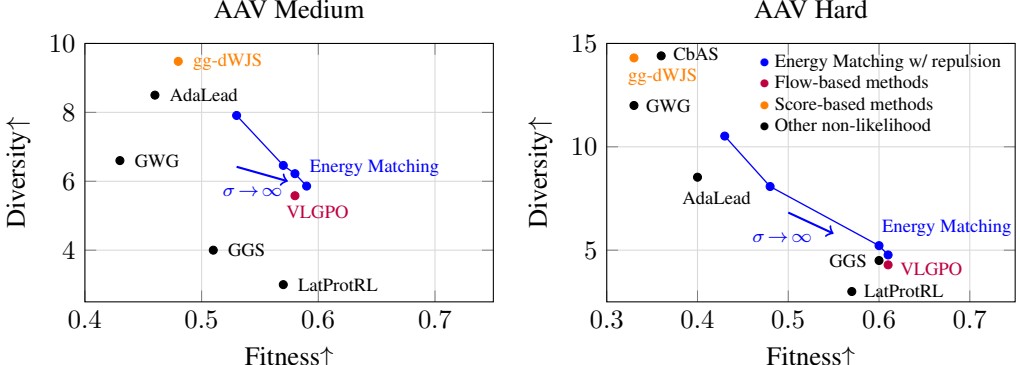

Figure 3: Fitness–diversity trade-off for protein inverse design on the AAV Medium (left) and Hard (right) benchmarks. We compare our Energy Matching method (blue), with diversity explicitly controlled by a repulsion strength parameter ($\propto \frac{1}{\sigma^2}$), against leading flow-based (purple), score-based (orange), and other non-likelihood methods (black). Fitness measures how well generated sequences satisfy the target property (predicted viral packaging efficiency), while diversity quantifies the average Levenshtein distance between sequences in each generated batch.

| Spearman's correlation ↑ | MNIST | CIFAR-10 |
|---|---|---|
| ESS [Johnsson et al., 2014] | 0.444 | 0.326 |
| FLIPD [Kamkari et al., 2024] | 0.837 | 0.819 |
| NB [Stanczuk et al., 2024] | 0.864 | 0.894 |
| Energy Matching (Ours) | 0.877 | 0.901 |

Table 3: Spearman's correlation coefficients of LID estimates with PNG compression rate. Benchmarks results reported in [Kamkari et al., 2024].

**Hessian-based LID Estimation.** Unlike diffusion models, EBMs explicitly parametrize the relative data likelihood. This explicit parametrization enables efficient analysis of the curvature of the underlying data manifold – in this example, estimating the LID. To this end, we compute the Hessian matrix $\nabla_x^2 V(x_{\text{data}})$ at a given data point and perform its spectral decomposition. We define near-zero eigenvalues as those whose absolute values lie within a small threshold $\tau$ (in our experiments, we set $\tau = 3$ for MNIST [Deng, 2012] and $\tau = 2$ for CIFAR-10). The count of near-zero eigenvalues reflects the number of *flat* directions and thus reveals the local dimension. As shown in Table 3, the LID estimates we obtain exhibit stronger correlations with PNG compression size[1] (evaluated on 4096 images) using Spearman's correlation. Figure 4 offers qualitative illustrations. Our EBM-based approach compares favorably to diffusion-based methods, as it relies on fewer approximations by performing computations exactly on the data manifold rather than merely in its vicinity.

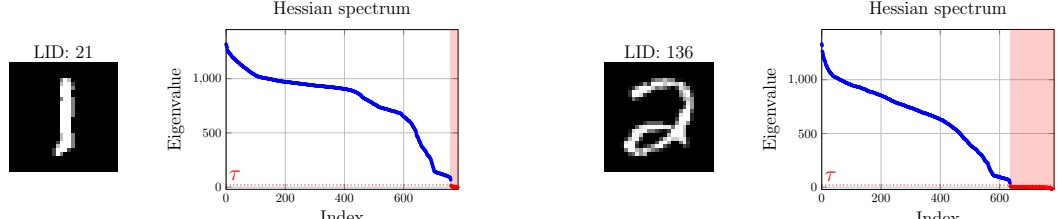

Figure 4: Qualitative results for LID estimation using the Hessian spectrum of $V_\theta(x)$. Left: Spectrum for a low-LID image. Right: Spectrum for a high-LID image. The eigenvalues quantify curvature along principal directions (eigenvectors). A degenerate spectrum (many near-zero eigenvalues, marked in red) indicates locally "flat" regions, revealing the LID. Intuitively, higher image complexity often corresponds to a higher LID.

---

[1]PNG is a lossless compression scheme specialized for images and can provide useful guidance when no LID ground truth is available Kamkari et al. [2024].

# 4 Conclusion and limitations

**Contributions.** We introduced a generative framework, *Energy Matching*, that reconciles the advantages of EBMs and OT flow matching models for simulation-free likelihood estimation and efficient high-fidelity generation. Specifically, it:

- Learns a *time-independent scalar potential energy* whose gradient drives rapid high-fidelity sampling–surpassing state-of-the-art energy-based models–while also forming a Boltzmann-like density well suitable for controlled generation. All without auxiliary generators.
- Offers efficient sampling from target data distributions on par with the state-of-the-art, while learning the score at the data manifold with manageable trainable parameters overhead.
- Offers a simulation-free, principled likelihood estimation framework for solving inverse problems—where additional priors can be easily introduced—and enables the estimation of a data point's LID with fewer approximations than score-based methods.

**Limitations.** First, our method requires an additional gradient computation with respect to the input, which can increase GPU memory usage (e.g., by 20–40%), particularly during training. Second, when estimating the LID (Section 3.3) for very high-dimensional datasets, computing the full Hessian spectrum may be impractical due to its computational complexity of $O(d^3)$; in such cases, partial-spectrum methods such as random projections or iterative solvers can be employed instead.

**Outlook.** Contrary to widespread belief, we demonstrated that time-independent irrotational methods for generative flows are highly effective and offer an exciting direction for future research. Our Energy Matching approach has the potential to yield novel insights into controlled generation and inverse problems for cancer research Weidner et al. [2024], Balcerak et al. [2024], molecules and proteins Wu et al. [2022], Bilodeau et al. [2022], computational fluid dynamics Gao et al. [2024], Shysheya et al. [2024], Molinaro et al. [2024], and other fields where precise control over generated samples and effective integration of priors or constraints are crucial. Moreover, Energy Matching aligns naturally with recent generative AI trends toward scaling inference for new capabilities [Zhang et al., 2025, Ma et al., 2025], further broadening its potential impact across scientific and engineering domains.

## Acknowledgments and Disclosure of Funding

This research was supported by the Helmut Horten Foundation and the European Cooperation in Science and Technology (COST).

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

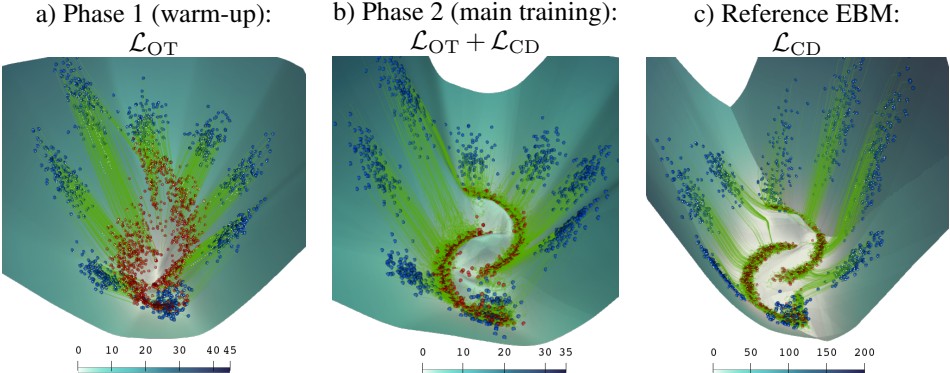

a) Phase 1 (warm-up): $\mathcal{L}_{\mathrm{OT}}$

b) Phase 2 (main training): $\mathcal{L}_{\mathrm{OT}} + \mathcal{L}_{\mathrm{CD}}$

c) Reference EBM: $\mathcal{L}_{\mathrm{CD}}$

Figure 5: Visualization of the energy $V_\theta(x)$ landscapes driving the samples from eight Gaussians to two moons. See Figure 1 for the 2D perspective. (a) The OT flow loss enforces zero curvature in $V_\theta(x)$ along the trajectories to the target. (b) Around the 2 Moons, the curvature of $V_\theta(x)$ is adjusted to approximate $\log p_{\mathrm{moons}}(x) \propto V_\theta(x)$ while remaining close to the pretrained landscape elsewhere. Combining these objectives yields a potential energy landscape that is both efficient for sampling and representative of the underlying target data distribution. (c) An EBM is shown for comparison, trained using contrastive divergence loss. Visible mode collapse that slows down the equilibration. Less regular landscape away from the data as it needs many simulations to explore it.

# A   Additional details on Energy Matching

In this section, we provide additional studies and visualizations on our method.

## A.1   Energy landscape during training

In Figure 5, we visualize how the potential $V_\theta(x)$ transitions from a flow-like regime, where the OT loss enforces nearly zero curvature away from the data manifold (a), to an EBM-like regime, where the curvature around the new data geometry (here, two moons) is adaptively increased to approximate $\log p_{\mathrm{data}}(x)$ (b). This two-stage design yields a well-shaped landscape that is both efficient to sample (thanks to a mostly flat potential between clusters) and accurate for density estimation near the data modes. For comparison, (c) shows an EBM trained solely with contrastive divergence, exhibiting sharper but less globally consistent basins.

## A.2   Ablation on the sampling time

Here, we provide ablation studies on CIFAR-10 unconditional generation. Specifically, we first pretrain using $\mathcal{L}_{\mathrm{OT}}$, and then fine-tune with $(\mathcal{L}_{\mathrm{OT}} + \mathcal{L}_{\mathrm{CD}})$, producing a stable Boltzmann distribution from which one can sample. Figure 6 illustrates the FID as a function of sampling time $\tau_{\mathrm{s}}$ for models trained under these different regimes. In the case of pure $\mathcal{L}_{\mathrm{OT}}$, the quality measure drops (FID increases) sharply when sampling at $\tau_{\mathrm{s}} > 1$; this occurs because, once the samples move close to the data manifold, there is no Boltzmann-like potential well to keep them from drifting away. Because the fidelity slope near the data manifold is steep with respect to sampling time, methods lacking explicit time-conditioning can easily overshoot or undershoot, significantly impacting fidelity. This behavior might explain why some models degrade in performance when made time-independent, as recently reported by Sun et al. [2025].

In Figure 6 we also report results for different values of the temperature-switching parameter $\tau^*$, which influences the sampling along the paths towards the data manifold (see Equation (3)).

## A.3   Ablations on the OT Solver

We evaluate the computational overhead and sensitivity to solver choice of the OT solver employed in our experiments.

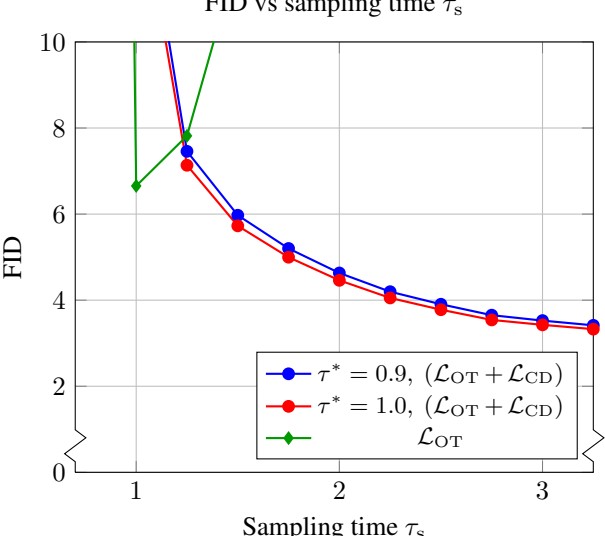

Figure 6: CIFAR-10 unconditional generation FID vs. sampling time $\tau_s$ when sampling from models trained under different scenarios: pure $\mathcal{L}_{OT}$ and combined ($\mathcal{L}_{OT} + \mathcal{L}_{CD}$), with temperature regime switching parameter $\tau^* \in \{0.9, 1.0\}$ during sampling. Lower FID indicates better generative quality. All results for Euler-Heun with $\Delta t = 0.01$.

**Computational overhead.** In the CIFAR-10 experiment, the OT solver accounts for roughly 1.5% of the training iteration time during Phase 1 Algorithm 1. This overhead decreases to a negligible level (approximately 0.01%) in Phase 2 Algorithm 2, where computational costs are predominantly dominated by the generation of negative samples.

**Impact of solver accuracy and complexity.** Let us (over-idealise) and model a standardised CIFAR-10 image as a vector $x \in \mathbb{R}^d$, drawn from $\mathcal{N}(0, I_d)$, with dimensionality $d = 32 \times 32 \times 3 = 3072$, and paired with an independent Gaussian-noise vector $z \sim \mathcal{N}(0, I_d)$. Each coordinate of the difference $(z - x)$ thus follows $\mathcal{N}(0, 2)$, and the squared Euclidean distance distribution is $|z - x|_2^2 \sim 2\chi_d^2$, which has mean $2d$, standard deviation $\sqrt{8d}$, and relative spread $\frac{\sqrt{8d}}{2d} \approx 0.025$. This demonstrates the "thin-shell" phenomenon, implying that all entries of the cost matrix $C_{ij} = |z_i - x_j|_2^2$ concentrate around nearly identical values, consistent with the distance-concentration effect observed by [Aggarwal et al., 2001]. Consequently, the choice among exact linear programming (LP), entropic regularisation (Sinkhorn), or even random matching should yield nearly identical cumulative optimal-transport costs, despite their different complexities: $O(n^3 \log n)$ for LP, $O(\frac{n^2}{\kappa})$ for Sinkhorn (with regularisation strength $\kappa$), and $O(n)$ for random matching.

Empirical evidence supporting this is summarised below:

- Our CIFAR-10 runs: FID degrades from 3.34 (LP) to 3.37 (random).
- [Tong et al., 2023] Table 5: FID scores of 4.44 (LP) vs 4.46 (random) at 100 steps.
- [Tong et al., 2023] Fig. D.2: solver accuracy saturates beyond batch size 16 in the 2D task.
- [Terpin et al., 2024] App. C.2: LP and Sinkhorn methods produce indistinguishable results unless regularisation is extreme.

We employ the LP approach for robustness, negligible cost, and no additional hyperparameters.

### A.4 Sampling Time Analysis

Computing the gradient $\nabla_x V_\theta(x)$ via automatic differentiation (`autograd` [Ansel et al., 2024]) introduces additional computational overhead compared to directly evaluating $V_\theta(x)$. Specifically, on the CIFAR-10 network architecture (see Figure 7), gradient evaluation is approximately $2.15\times$

slower, as it requires both forward and backward passes. In contrast, flow matching and diffusion models directly parameterize the velocity field, thus only requiring forward computations during sampling.

Nevertheless, despite this per-step computational cost, our method achieves competitive overall sampling efficiency due to a reduced number of integration steps needed for high-quality generation. As demonstrated in Table 4, Energy Matching achieves a lower FID (3.34) in 173 seconds per batch, outperforming OT-FM (FID 3.74 in 136 seconds per batch) and DDPM++ (FID 3.45 in 183 seconds per batch). Our results thus indicate a favorable balance between computational overhead per step and total sampling runtime.

Table 4: Comparison of sampling efficiency and quality on CIFAR-10 (batch size 128, NVIDIA R6000 48GB GPU). Despite gradient computation overhead ($\nabla_x V_\theta(x)$ via backward pass), Energy Matching achieves superior FID scores with competitive wall-clock sampling time.

| Method | Params | Steps | Sampling Time [s]↓ | FID↓ |
|---|---|---|---|---|
| **Flow-/Diffusion-based Models** | | | | |
| OT-FM [Tong et al., 2023] | 37M | 1000 | **136** | 3.74 |
| DDPM++ [Kim et al., 2021] | 62M | 1000 | 183 | 3.45 |
| **Energy-based Models** | | | | |
| Energy Matching *(Ours)* | 50M | 325 | 173 | **3.34** |

# B    Details on AAV inverse design protein generation

We optimize protein fitness for adeno-associated virus (AAV) sequences in the *medium* and *hard* data regimes proposed by [Kirjner et al., 2024], using latent encodings and backbone architectures from [Bogensperger et al., 2025]. Conditional sampling employs classifier guidance via learned predictor networks $g_\phi$ or $\tilde{g}_\phi$ to steer samples toward high-fitness regions. The CNN-based fitness predictors from [Kirjner et al., 2024] are trained only on the limited training data for each regime.

Training follows Algorithm 1 and Algorithm 2. We sample 128 sequences using Algorithm 3, keeping the same batch size across all baselines; detailed hyperparameters are given in Section D. Generated sequences are evaluated for fitness using the learned oracle from [Kirjner et al., 2024], and further assessed for both intra-set diversity and novelty relative to the training sequences [Jain et al., 2022]. Our approach achieves state-of-the-art fitness while improving diversity (see Table 5). Incorporating interaction energy in Algorithm 3 further enhances diversity with manageable impact on fitness.

Table 5: AAV optimization results. For VLGPO (flow-based) and Energy Matching, medium difficulty uses $g_\phi$, hard difficulty uses $\tilde{g}_\phi$. Metrics (*Fitness*↑, *Diversity*↑, *Novelty*↑). Reported uncertainty of Fitness is expressed as standard deviation.

| Method | AAV medium | | | AAV hard | | |
|---|---|---|---|---|---|---|
| | Fitness↑ | Diversity↑ | Novelty↑ | Fitness↑ | Diversity↑ | Novelty↑ |
| **Learning Unnormalized Data Likelihood** | | | | | | |
| **Energy-based Models** | | | | | | |
| Energy Matching *(Ours)* | 0.59 (0.0) | 5.86 | 5.0 | 0.61 (0.0) | 4.77 | 6.7 |
| Energy Matching (+repulsion) *(Ours)* | 0.58 (0.0) | 6.22 | 5.0 | 0.60 (0.0) | 5.22 | 6.6 |
| **Learning Transport/Score Along Noised Trajectories** | | | | | | |
| **Flow-based Models** | | | | | | |
| VLGPO [Bogensperger et al., 2025] | 0.58 (0.0) | 5.58 | 5.0 | 0.61 (0.0) | 4.29 | 6.2 |
| **Diffusion Models** | | | | | | |
| gg-dWJS [Ikram et al., 2024] | 0.48 (0.0) | 9.48 | 4.2 | 0.33 (0.0) | 14.3 | 5.3 |
| **Other Methods** | | | | | | |
| LatProtRL [Lee et al., 2024] | 0.57 (0.0) | 3.00 | 5.0 | 0.57 (0.0) | 3.00 | 5.0 |
| GGS [Kirjner et al., 2024] | 0.51 (0.0) | 4.00 | 5.4 | 0.60 (0.0) | 4.50 | 7.0 |
| AdaLead [Sinai et al., 2020] | 0.46 (0.0) | 8.50 | 2.8 | 0.40 (0.0) | 8.53 | 3.4 |
| CbAS [Brookes et al., 2019] | 0.43 (0.0) | 12.70 | 7.2 | 0.36 (0.0) | 14.4 | 8.6 |
| GWG [Grathwohl et al., 2021] | 0.43 (0.1) | 6.60 | 7.7 | 0.33 (0.0) | 12.0 | 12.2 |
| GFN-AL [Jain et al., 2022] | 0.20 (0.1) | 9.60 | 19.4 | 0.10 (0.1) | 11.6 | 19.6 |

## C  Details on LID estimation

**Definition.**    To start, we need to introduce the concept of *local mass*, defined as

$$M(r) \;=\; \int_{B(x_{\text{data}};r)} p(x)\,dx,$$

where $p(x)$ is the local density and $B(x_{\text{data}}, r)$ is a ball of radius $r$ around $x_{\text{data}}$, i.e. $B(x_{\text{data}}, r) = \{x \in \mathbb{R}^d : \|x - x_{\text{data}}\| \le r\}$. The LID is then given by:

$$\text{LID}(x_{\text{data}}) = d - \lim_{r \to 0} \frac{\log\big(M(r)\big)}{\log(r)}.$$

Intuitively, $M(r)$ measures how much probability mass is concentrated in a ball of radius $r$ around $x_{\text{data}}$. As we shrink this ball, the growth rate of $M(r)$ in terms of $r$ reveals the local dimensional structure of the data.

**Assumptions.**    In the context of contrastive divergence, we assume that data points $x_{\text{data}}$ lie in well-like regions [Hyvärinen, 2006], i.e.:

$$\nabla V(x_{\text{data}}) \approx 0 \quad \text{and} \quad \nabla^2 V(x_{\text{data}}) \text{ is positive semidefinite (or nearly so).}$$

Conceptually, $V(x)$ can be thought of as an energy function; points where $\nabla V(x_{\text{data}}) = 0$ are near local minima of this energy, and the Hessian $\nabla^2 V(x_{\text{data}})$ provides information about local curvature (see Figure 4 for a qualitative illustration).

**Energy-based density.**    We define an energy-based density

$$p(x) \;\propto\; \exp\!\Big(-\tfrac{V(x)}{\varepsilon}\Big),$$

where $\varepsilon$ is a temperature parameter. Near a data point $x_{\text{data}}$ satisfying $\nabla_x V(x_{\text{data}}) = 0$, we can approximate $V(x)$ by its second-order Taylor expansion:

$$V(x) \approx V(x_{\text{data}}) + \frac{1}{2}(x - x_{\text{data}})^\top \nabla_x^2 V(x_{\text{data}})(x - x_{\text{data}}).$$

Consequently, in view of the assumptions above,

$$p(x) \propto \exp\left(-\frac{1}{2\varepsilon}(x - x_{\text{data}})^\top \nabla_x^2 V(x_{\text{data}})(x - x_{\text{data}})\right).$$

**Local mass derivation and the rank of the energy Hessian.**    Substituting the local quadratic form of $p(x)$ near $x_{\text{data}}$ into the definition of the local mass $M(r)$, we obtain:

$$M(r) = \int_{B(x_{\text{data}},r)} p(x)dx \propto \int_{B(x_{\text{data}},r)} \exp\left(-\tfrac{1}{2\varepsilon}(x - x_{\text{data}})^\top \nabla_x^2 V(x_{\text{data}})(x - x_{\text{data}})\right) dx.$$

For small $r$, the dominant contribution depends on the rank of the Hessian $\nabla_x^2 V(x_{\text{data}})$. Let $k = \text{rank}(\nabla_x^2 V(x_{\text{data}}))$. Then, as $r \to 0$, one can show that $M(r) = Cr^k$, where $C$ does not depend on $r$. We take the logarithm on both sides and divide by $\log(r)$ to get

$$\frac{\log(M(r))}{\log(r)} = \frac{\log(C) + k\log(r)}{\log(r)} = k + \frac{\log(C)}{\log(r)},$$

and the second term vanishes as $r \to 0$. Hence,

$$\text{LID}(x_{\text{data}}) = d - k.$$

**Practical estimation.**    In practice, the LID at a data point $x_{\text{data}}$ can be estimated through the following procedure:

1. Train $V(x)$ with Energy Matching.
2. Compute the Hessian $H = \nabla_x^2 V(x_{\text{data}})$.
3. Perform an eigenvalue decomposition on $H$.

Then the estimated local data-manifold dimension corresponds to the number of directions with negligible curvature (smaller magnitude than some $\tau$).

# D   Training details

Below, we detail the training configurations for CIFAR-10, ImageNet 32x32, CelebA, MNIST, and AAV. Additionally, we provide intuitions for practical hyperparameter choices to facilitate effective training across additional datasets. We recommend using SiLU activation functions wherever possible, as they smooth out the energy landscape and improve the numerical stability of the $\nabla_x V(x)$ computation. The gradient of the potential, $\nabla_x V(x)$, is computed using automatic differentiation via PyTorch's `autograd` [Ansel et al., 2024]. We optimize all models using the Adam optimizer [Kingma and Ba, 2014] and maintain an exponential moving average (EMA) of the model weights.

While we specifically adopt (i) a one-sided trimmed mean of negative sample energies and (ii) clamping of the contrastive loss for stability, any commonly used EBM technique (e.g., persistent contrastive divergence [Tieleman, 2008], replay buffers, multi-scale negative sampling) could be readily employed.

In our approach, we introduce two hyperparameters, $\alpha$ and $\beta$, to control these stabilizing techniques:

$\alpha$ = fraction of negative energies discarded to remove outliers that skew the mean (e.g., top 10%),

$\beta$ = clamp threshold for $\mathcal{L}_{\mathrm{CD}}$ (i.e., we clamp $\mathcal{L}_{\mathrm{CD}}$ to be $\geq -\beta$).

**CIFAR-10:**   The architecture is shown in Figure 7. We use the same UNet from [Tong et al., 2023] (with fixed $t = 0.0$, making it effectively time-independent) followed by a small vision transformer (ViT) [Dosovitskiy et al., 2020] to obtain a scalar output. Hyperparameters are: $\tau_s = 3.25$, $\tau^* = 1.0$, $\Delta t = 0.01$, $M_{\mathrm{Langevin}} = 200$. We train for 145k iterations using Algorithm 1 with EMA 0.9999 and then 2k more with Algorithm 2 and EMA 0.99 on 4xA100. The batch size is 128, learning rate is $1.2 \times 10^{-3}$, $\varepsilon_{\max} = 0.01$, $\lambda_{\mathrm{CD}} = 1 \times 10^{-3}$, $\alpha = 0.1$, and $\beta = 0.02$. Negatives initialized on the data manifold follow the same temperature schedule as those initialized from the noise.

**ImageNet 32x32:**   The architecture is shown in Figure 7 (same as for CIFAR-10). Hyperparameters are: $\tau_s = 2.5$, $\tau^* = 1.0$, $\Delta t = 0.01$, $M_{\mathrm{Langevin}} = 200$. We train for 640k iterations using Algorithm 1 with EMA 0.9999 and then 1k more with Algorithm 2 and EMA 0.99 on 7xA100. The batch size is 128, learning rate is $6 \times 10^{-4}$, $\varepsilon_{\max} = 0.01$, $\lambda_{\mathrm{CD}} = 1 \times 10^{-3}$, $\alpha = 0.1$, and $\beta = 0.02$. Negatives initialized on the data manifold follow the same temperature schedule as those initialized from the noise.

**CelebA:**   We scale the CIFAR-10 model by $\sim 2\times$; see Figure 8. We set $\tau_s = 2.0$, $\tau^* = 1.0$, $\Delta t = 0.01$, $M_{\mathrm{Langevin}} = 200$, and train for 250k iterations using Algorithm 1 with EMA 0.9999 then 4k with Algorithm 2 and EMA 0.99 on 4xA100. The batch size is 32, learning rate is $1 \times 10^{-4}$, $\varepsilon_{\max} = 0.05$, $\lambda_{\mathrm{CD}} = 1 \times 10^{-4}$.

**MNIST:**   We downscale the CIFAR-10 model (Figure 7) to 2M parameters by reducing the UNet base width to 32 channels, using channel multipliers [1, 2, 2], setting the number of attention heads in the UNet to 2, simplifying the Transformer head to an embedding dimension of 128, 2 Transformer layers, 2 attention heads, and adjusting the output scale to 100.0. We set $\tau_s = 2.0$, $\tau^* = 1.0$, $\Delta t = 0.025$, $M_{\mathrm{Langevin}} = 75$, and train for 50k iterations using Algorithm 1 with EMA 0.999 then 3.3k with Algorithm 2 and EMA 0.99 on a single A100. The batch size is 128, learning rate is $1 \times 10^{-4}$, $\varepsilon_{\max} = 0.1$, $\lambda_{\mathrm{CD}} = 1 \times 10^{-3}$, $\alpha = 0.0$, and $\beta = 0.05$. Negatives initialized on the data manifold follow the same temperature schedule as those initialized from the noise.

**AAV:**   We adopt the one-dimensional CNN architecture as used in [Bogensperger et al., 2025], summing the final-layer activations to obtain the potential. We train for 10k iterations using Algorithm 1 and for 1k iterations using Algorithm 2 on a single A100. The batch size is 128, learning rate is $1 \times 10^{-4}$, $\varepsilon_{\max} = 0.1$, $M_{\mathrm{Langevin}} = 200$, $\Delta t = 0.01$, and $\lambda_{\mathrm{CD}} = 1 \times 10^{-4}$. For Algorithm 3 we use $\tau_s = 1.7$ for AAV medium and $\tau_s = 1.3$ for AAV hard, $\tau^* = 0.9$, $\zeta = 0.01$ for AAV medium and $\zeta = 0.009$ for AAV hard. We set the target fitness to $y = 1$ to aim for the maximum fitness in the generated sequences.

**Intuition for other datasets:**   It is essential for negative samples to reach the equilibrium distribution induced by the model or at least the proximity of the data manifold. The condition $M_{\mathrm{Langevin}} \times \Delta t \gg 1$ is critical to achieving this, with $\Delta t$ small enough to ensure that negative samples remain of sufficient

quality—typically the same $\Delta t$ as used in flow matching for the given generation task. In practice, we set $M_{\text{Langevin}} \times \Delta t = 2$ across most experiments. We recommend starting with $\tau^* = 1.0$ to ensure optimal transport regularization near the data manifold, thereby enhancing training stability. If additional conditions are required during sampling, exploring lower values ($\tau^* < 1.0$) may be beneficial, as this parameter does not need to remain consistent between training and sampling (as shown in Figure 6). Training with $\tau^* < 1.0$ is advised only in special cases, such as extremely low-dimensional problems like that shown in Figure 1, where it is possible to simultaneously be far from the data manifold (from the perspective of the target mode) and close to it (from the perspective of another mode). The parameter $\varepsilon_{\text{max}}$ controls how extensively negative samples explore the space. For unconditional generation, we use $\varepsilon_{\text{max}} = 0.01$, but for inverse problems or design tasks, higher values (e.g., $\varepsilon_{\text{max}} = 0.05$) can improve robustness. The parameter $\tau_s$ significantly depends on the task (unconditional or conditional) and thus must be tuned accordingly post-training. Without explicit tuning, selecting $\tau_s = M_{\text{Langevin}} \times \Delta t$ is a reasonable default. Finally, parameters $\lambda_{\text{CD}}$, $\alpha$, and $\beta$ influence the stability of contrastive training and must be empirically determined alongside appropriate early stopping.

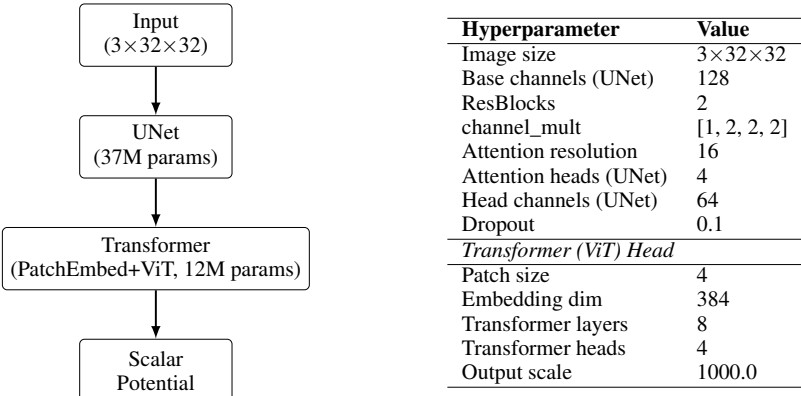

| Hyperparameter | Value |
|---|---|
| Image size | 3×32×32 |
| Base channels (UNet) | 128 |
| ResBlocks | 2 |
| channel_mult | [1, 2, 2, 2] |
| Attention resolution | 16 |
| Attention heads (UNet) | 4 |
| Head channels (UNet) | 64 |
| Dropout | 0.1 |
| *Transformer (ViT) Head* | |
| Patch size | 4 |
| Embedding dim | 384 |
| Transformer layers | 8 |
| Transformer heads | 4 |
| Output scale | 1000.0 |

Figure 7: Diagram of our UNet+Transformer EBM for CIFAR-10 and ImageNet 32x32. A UNet (37M params) processes a 3×32×32 image; its output is fed into a Transformer head (PatchEmbed + 8-layer ViT, 12M params) that produces a scalar potential. Here we employ the identical UNet architecture as in [Tong et al., 2023], but with the time parameter fixed at $t = 0$ to render the model time-independent.

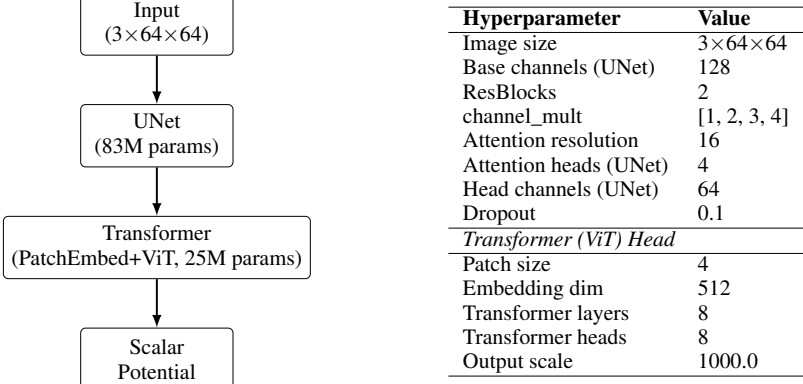

| Hyperparameter | Value |
|---|---|
| Image size | 3×64×64 |
| Base channels (UNet) | 128 |
| ResBlocks | 2 |
| channel_mult | [1, 2, 3, 4] |
| Attention resolution | 16 |
| Attention heads (UNet) | 4 |
| Head channels (UNet) | 64 |
| Dropout | 0.1 |
| *Transformer (ViT) Head* | |
| Patch size | 4 |
| Embedding dim | 512 |
| Transformer layers | 8 |
| Transformer heads | 8 |
| Output scale | 1000.0 |

Figure 8: Diagram of our UNet+Transformer EBM for CelebA. A UNet (83M params) processes a 3×64×64 image; its output is fed into a Transformer head (PatchEmbed+8-layer ViT, 25M params) that produces a scalar potential.

