# OpenReview forum: "Energy Matching: Unifying Flow Matching and Energy-Based Models for Generative Modeling"
_NeurIPS.cc/2025/Conference — NeurIPS 2025 poster_

### Official Review · Reviewer_FNAJ · 2025-06-21

**Clarity:** 4
**Significance:** 3
**Originality:** 3
**Rating:** 5
**Confidence:** 4

**Summary:**

The manuscript proposes a hybrid generative model formulation that combines flow matching and energy-based modeling. The core concept behind such formulation is the Jordan–Kinderlehrer–Otto (JKO) scheme that selects a probability distribution in the next timestep as minimiser of linear combination of: Wasserstein distance between the target distribution and the distribution in the current timestep, potential energy with parameterized scalar potential, and internal energy expressed as negative entropy tuned with a temperature-like parameter $\epsilon(t)$. Following previous work [Terpin et al., 2024], the JKO scheme is analysed under its first-order optimality conditions at each timestep $t$ (defined with Eq. 4). Different from previous works, the manuscript studies the resulting formulation depending on the time parameter $t$ and reveals two distinct regimes. In the case of early timesteps ($\epsilon(t)$ \approx 0), Eq. 4 approximately behaves like a flow-matching OT with time-independent potential. Such straight paths are useful for fast space traversal far from the data manifold. In the case of later timesteps nearer to the data manifold ($\epsilon(t)$ \approx 1), the scalar potential can be reformulated into Boltzmann's energy. Thus, the sampling trajectories are allowed to curve, which adds to the model's expressivity and matching of the data manifold. The parameterised scalar potential is trained in two stages. Warmup stage optimises the OT objective characteristic for flow matching, while the main training stage combines the OT objective with a contrastive objective typical for EBMs. Experimental evaluation indicates strong performance in unconditional image generation, inverse problems, and  estimation of local intrinsic dimension.

**Questions:**

See weaknesses.

**Ethical Concerns:**

["NO or VERY MINOR ethics concerns only"]

**Final Justification:**

The authors presented compelling answers to all raised concerns. Thus, I recommend acceptance.

**Limitations:**

Yes

**Quality:**

4

**Strengths And Weaknesses:**

**Strengths:**

S1. The proposed hybrid generative formulation elegantly combines OT-based flow matching and EBMs through time-dependent analysis.

S2. The manuscript is well written and easy to follow.

S3. Experimental evaluation indicates competitive performance for three different applications.

**Weaknesses:**

W1. The assumption $\rho_t \approx \rho_{eq}$ (Line 84) may not hold in practice, especially early in the training. The manuscript should elaborate more on the feasibility of such an assumption. Perhaps visualise contributions of the tree components in Eq. 4 depending on $t$. My intuition is that this is why the warmup stage is needed in the first place.

W2. The manuscript reports only the FID score for unconditional image generation. Reporting precision and recall [a] would further strengthen the results.

W3. The proposed training procedure (Algorithm 2) increases training time compared to the standard flow matching due to the sampling required by the contrastive objective. This should be quantified and discussed in ablations (and possibly added to limitations).

W4. The manuscript parameterises the scalar potential $V_{\theta}$ as a deep model. This implies that both forward and backward passes are needed for sampling (as in EBMs). This is significantly slower than in the case of direct parameterisation of vector fields $\nabla_x V_{\theta}$  (typicall for flow matching/score matching). The manuscript should discuss this. For example present wallclock time vs FID graph for different baselines.

[a] Tuomas Kynkäänniemi, Tero Karras, Samuli Laine, Jaakko Lehtinen, Timo Aila: Improved Precision and Recall Metric for Assessing Generative Models. NeurIPS 2019

---

> ### Author Rebuttal · Authors · 2025-07-30
>
> We thank the reviewer for their thoughtful comments and helpful suggestions, which allow us to clarify important aspects of our manuscript. Below, we provide detailed responses addressing the raised weaknesses (W) and questions (Q):
>
> **W1: The assumption  $p_t \approx p_\{eq}$ (Line 84) may not hold in practice, especially early in the training. The manuscript should elaborate more on the feasibility of such an assumption. Perhaps visualise contributions of the tree components in Eq. 4 depending on $t$. My intuition is that this is why the warmup stage is needed in the first place.**
>
> We thank the reviewer for highlighting this important point, as it reveals a missed opportunity in our manuscript to clearly explain the motivation behind the warm-up phase.
> The statement from line 84 that $p_t$ is close to $p_{eq}$ (in the context of $p_t$ being ‘near data manifold’) indeed may not hold in practice, particularly in the early stages of training. However, line 84 does not imply any guarantees for untrained $V(x)$. Instead, the subsection "First-order optimality conditions" (pp. 3–4) describes properties of the desired solution (such as $p_{data} = p_{eq}$ and Eq. 4) rather than assumptions that must hold throughout training.
> During training, we explicitly measure how far the current distribution $p_t$ deviates from these optimal conditions and use this imbalance as our learning signal.
>
> We welcome the reviewer's suggestion to visualize how the three forces evolve during training. We will include this visualization in supplementary material.
>
> Early in training, when the potential $V$ is still untrained, contrastive terms provide limited guidance because the negative sample distribution $p_t$ may either be too distant from the data manifold (if initialized in noise) or too close due to poor mixing (if initialized directly on the manifold). This perspective clearly highlights the necessity of the warm-up phase identified by the reviewer. As briefly mentioned around lines 143–144, it is crucial to have negative samples properly distributed around the data manifold to probe low-density regions, thereby constructing energy walls that clearly separate these regions from the high-density data manifold. Once the warm-up phase sufficiently aligns $p_t$ with $p_{data}$, the contrastive terms can more effectively enforce the condition $p_t \approx p_{data} \Leftrightarrow p_t \approx p_{eq}$, actively pulling $p_{eq}$ closer to $p_{data}$.
>
> We thank the reviewer and will explicitly add a sentence at the start of the "First-order optimality conditions" subsection, clarifying that these conditions represent the optimization goal (features of the solution) and highlighting the critical role of the warm-up phase.
>
>
> **W2: The manuscript reports only the FID score for unconditional image generation. Reporting precision and recall [a] would further strengthen the results.**
>
> We thank the reviewer for this valuable suggestion. While our results measured by FID significantly surpass existing EBMs (halving the best reported FID scores), we acknowledge that FID alone provides a limited perspective. For proteins, we indeed report additional metrics such as diversity and novelty alongside fidelity measures (see Table 4 in Appendix B, "Posterior sampling"). In the revised manuscript, we will explicitly highlight that, given our FID scores are now on par with state-of-the-art methods, future works on image generation with EBMs should incorporate complementary metrics such as precision and recall [a] for a more complete evaluation.
>
> **W3: The proposed training procedure (Algorithm 2) increases training time compared to the standard flow matching due to the sampling required by the contrastive objective. This should be quantified and discussed in ablations (and possibly added to limitations).**
>
> While the proposed training procedure indeed adds computational overhead compared to standard flow matching due to the contrastive sampling, we view this overhead as justified given the additional capabilities provided by our approach. As illustrated clearly in Figure 5a, the OT phase alone (without contrastive sampling) achieves results similar to flow matching (same objective, different parameterization), and does not require sampling negative examples. The contrastive objective refines the learned landscape, explicitly enabling functionalities that standard flow-based models cannot achieve directly, such as evaluating log-likelihoods in a single forward pass and performing posterior sampling under arbitrary conditions. Thus, if these functionalities are desired, the additional computation cannot be seen purely as overhead but as necessary to enable these key features.
>
> We fully agree that this computational cost should be clearly quantified and discussed. To address this, we provide the following timing breakdown (per batch on CIFAR-10 using a single NVIDIA R6000):
>
> | Batch | Phase | Total [s] | OT coupling [s] (%) | Flow loss [s] (%) | CD loss [s] (%) |
> |-------|-------|-----------|---------------------|-------------------|-----------------|
> | 128   | 1     | 0.260    | 0.004 (1.5%)      | 0.256 (98.5%)   | 0 (0%)          |
> | 128   | 2     | 33.030   | 0.004 (0.01%)     | 0.270 (0.82%)   | 32.755 (99.17%)|
>
> **Additional Timings:** EM potential evaluation: **0.148 s**; EM velocity generation: **0.317 s**
>
>
> In short, the flow component is roughly 2x slower compared to directly training a velocity field via flow matching due to the additional backward pass. While the CD loss computation itself is approximately 100x slower than the flow loss, CD iterations constitute only about 1–5% of total flow iterations (see Training Details). Using our current negative-sample generation approach (no replay buffers, small fixed step integration), the total training time can increase by up to 12 times in the extreme cases, due to both the curl-free parametrization and negative-sample simulations.
>
> Although this overhead remains negligible for shorter tasks (e.g., protein generation, small images), it could become significant in larger settings. Future improvements could incorporate techniques such as persistent contrastive divergence [E1, E2], replay buffers, or adaptive Langevin steps to significantly reduce generation costs of the negative samples. We omitted these common engineering tricks here in favor of clarity of presentation.
>
> We thank the reviewer for highlighting this point. The overhead described here is characteristic of most traditional EBMs, which typically rely on simulating negative samples during training. Although simulation-based training lost popularity in recent years due to its computational demands and the rise of simulation-free approaches, improvements in hardware capabilities are now bringing simulated negatives back into practical relevance, as they provide complementary signals to simulation-free supervision. Importantly, this overhead is incurred mainly during training and the method remains competitive in terms of inference speed (addressed further in **W4**). Additionally, Energy Matching can naturally extend to discrete and structured domains, such as molecule generation, where recent work [E3] explores replacing Langevin-based MCMC negative sampling with analytical solutions derived from heat equations on structured spaces.
>
> We will clearly emphasize this context in both the manuscript (Introduction) and appendix (Training Details), making it accessible especially to readers less familiar with EBMs.
> - [E1] T. Tieleman and G. Hinton, “Using Fast Weights to Improve Persistent Contrastive Divergence” (ICML 2009)
> - [E2] C. Jarzynski, “Nonequilibrium equality for free energy differences”
> - [E3] T. Schröder et al., “Energy-Based Modelling for Discrete and Mixed Data via Heat Equations on Structured Spaces” (NeurIPS 2024)
>
> **W4: The manuscript parameterises the scalar potential  as a deep model. This implies that both forward and backward passes are needed for sampling (as in EBMs). This is significantly slower than in the case of direct parameterisation of vector fields  (typicall for flow matching/score matching). The manuscript should discuss this. For example present wallclock time vs FID graph for different baselines.**
>
> We fully agree that directly reporting parameter counts and sampling steps serves only as a proxy, and that explicitly comparing wall-clock times against achieved FID scores would offer clearer insights. Indeed, our scalar-potential parameterization requires both forward and backward passes for sampling, similar to traditional EBMs, which roughly doubles (as seen in **W3**)  sampling steps time compared to directly parameterizing vector fields.
>
> We sincerely thank the reviewer for emphasizing this important point and will include a clear wall-clock time vs. FID comparison plot in the revised manuscript. This visualization will clearly compare our method with baseline methods we successfully reproduced. Below, we present the measurements obtained on CIFAR-10 (batch size of 128, NVIDIA R6000), using Euler-Heun (325 steps) and Euler-Maruyama (1000 steps):
>
> | Method           | Params | Steps | Sampling time [s] ↓ | FID ↓               |
> |------------------|--------|-------|---------------------|---------------------|
> | OT-FM            | 37M    | 325   | **88**              | 4.04                |
> | DDPM++           | 62M    | 1000  | 183                 | 3.74                |
> | OT-FM            | 37M    | 1000  | 136                 | 3.45                |
> | Energy Matching  | 50M    | 325   | 173                 | **3.34**$^\dagger$  |
>
> - $\dagger$ *FID 3.34 achieved by Energy Matching differs from the original submission (FID 3.97) due to adjusted hyperparameters: $\tau^\*=1.0$ (vs $0.9$), $\lambda_{cd}=1\times10^{-3}$ (vs $8\times10^{-3}$), and 325 Langevin steps (vs 300). The architecture is unchanged; updated result in camera-ready.*

---

> > ### Comment · Reviewer_FNAJ · 2025-08-03
> > **Post-rebuttal**
> >
> > I thank the authors for their constructive response. I decide to keep my score (Accept) and hope to see parts of the presented response in the next manuscript version.

---

> > > ### Author Response · Authors · 2025-08-03
> > >
> > > We sincerely thank the reviewer for the constructive comments and for maintaining the accept recommendation. The reviewer’s suggestions have significantly improved the clarity and quality of our manuscript, and we will incorporate these points into the revised version.

---

### Official Review · Reviewer_Rsu1 · 2025-06-30

**Clarity:** 2
**Significance:** 3
**Originality:** 3
**Rating:** 5
**Confidence:** 2

**Summary:**

The paper "Energy Matching: ..." introduces a method that combines energy based models with flow type models using optimal transport theory. By doing that they can learn a single time invariant scalar field that can be used for unconditional sampling, inverse problems or local intrinsic dimension estimation. Results demonstrate strong performance, sometimes only within energy based models.

**Questions:**

- Please improve the general accessibility of the material to readers outside the OT field.
- The exposition of the results could be clearer. You write that you surpass state-of-the-art EBMs. This is technically true, but somehow omits the fact that other methods in the comparison achieve even better results. I am not in favor of only accepting results that beat state of the art, but you need to argue why your results are still relevant and what advantage/disadvantage other methods have compared to yours.

**Ethical Concerns:**

["NO or VERY MINOR ethics concerns only"]

**Final Justification:**

In light of the response of the authors to my and other questions, I believe it's a valuable and interesting contribution to NeurIPS. Regarding my review: The authors clarified the unifying aspect and put the results to non-EBM into context. I thus raised my score.

**Limitations:**

Are discussed in section 4.

**Quality:**

3

**Strengths And Weaknesses:**

Strength:
- The motivation of learning an energy based model is not time dependent, allowing for an efficient sampling scheme as well as inverse problems is important. The authors seem to make a relevant contribution to this question.

Weaknesses
- The exposition is very dense and barely understandable for someone outside the OT field.
- The paper title seems overstated. I does not read like a unifying theory of flow matching and energy based models, but rather a combination of the two.
- The experiments are limited to smaller scale data (imagenet 32x32) and its not clear how the methods scales. The authors do acknowledge that in the limitations.

---

> ### Author Rebuttal · Authors · 2025-07-30
>
> We thank the reviewer for their constructive suggestions. We provide point-by-point responses to the raised weaknesses (W) and questions (Q).
>
> **W1/Q1: The exposition is very dense and barely understandable for someone outside the OT field. / Please improve the general accessibility of the material to readers outside the OT field.**
>
> Thank you for raising this concern. We fully agree that providing intuitive explanations is important, especially for readers less familiar with optimal transport.
> In particular, we will add a concise paragraph clearly emphasizing the following key insight: a central OT goal in our method is to produce a potential landscape with minimal curvature—effectively a clean, gently sloped "path"—between the noise and data distributions. This smooth slope makes it easier and faster to generate high-quality samples using only a few ODE or SDE integration steps. This capability is essential, as generating accurate, high-quality samples allows us to produce reliable negative examples. These negative examples are then used by the contrastive objective to better capture the geometric structure, or "curvature," of the data manifold close to the actual data points. Thus, away from the data manifold, the OT component naturally regularizes the contrastive objective, enhancing both training stability and final generation quality.
>
> We will incorporate this intuitive perspective clearly in the method section before formally introducing the solution via the JKO scheme. Additionally, we will include a brief summary of essential OT concepts in the appendix for quick reference. We welcome any further suggestions to improve clarity.
>
>
>
> **W2: The paper title seems overstated. I does not read like a unifying theory of flow matching and energy based models, but rather a combination of the two.**
>
> Thank you for this valuable remark. We would like to clarify why we chose this specific title and agree that this deserves a clearer explanation. We will incorporate the following rationale explicitly into the camera-ready version to better communicate why we believe Energy Matching unifies flow matching and energy-based models.
> Our work is framed entirely in the JKO variational picture: one free‑energy functional is minimised at every step, and the temperature parameter $\varepsilon$ decides which regime we are in. For any positive $\varepsilon$ the updates coincide with the dynamics used to train energy‑based models; as $\varepsilon$ approaches zero the very same updates degenerate to the displacement‑only objective that flow matching employs.  We are, to the best of our knowledge, the first to show explicitly that flow matching is the zero‑temperature member of this JKO family, and the title is meant to foreground that link.
>
> Examining Algorithm 3 (p. 15) (also discussed in our response to **Q2** from reviewer **XTSh**) reveals a key limitation of flow matching which operates in the $\varepsilon=0$ limit. Every term that would steer the sampler toward a specific observation is multiplied by $\varepsilon$; setting $\varepsilon$ to zero (precisely the flow‑matching limit) makes those terms disappear. The model is then locked into a single transport path dictated by the prior and cannot produce samples from a conditioned posterior without outside tricks and approximations. Highlighting this drawback, and the way our framework overcomes it, is a novel contribution of the paper.
>
> The optimal‑transport portion of the loss provides positive guidance on where mass should move, while the contrastive-divergence loss contributes a negative signal that discourages excursions into low‑density regions. Rather than merely stitching two concepts together we show that they are part of the same variational picture with complementary losses for different regions of the trajectory.
>
> We agree that explicitly clarifying this rationale will strengthen our contribution and help prevent potential confusion regarding the title. To address your valuable feedback, we will prominently emphasize this unified variational interpretation in the abstract and introduction.
>
> **W3: The experiments are limited to smaller scale data (imagenet 32x32) and it's not clear how the methods scales. The authors do acknowledge that in the limitations.**
>
> Thank you for highlighting this important point regarding scalability. We agree that our current experiments focus primarily on image datasets of moderate resolution. This scale, however, is a widely studied benchmark, making it well-suited for demonstrating clearly the strengths and relative performance of our approach. Indeed, our method significantly surpasses the previous state-of-the-art FID scores for EBMs by achieving more than a two-fold improvement. We believe this demonstrates the potential to extend our method to higher-dimensional images in future work.
> Importantly, our approach is also well-suited to problems outside image generation, such as protein design, where typical data dimensionality is substantially lower than even moderate-resolution images. This is a significant and practically relevant domain that benefits directly from the explicit scalar-potential structure inherent in our approach, as demonstrated in our protein-generation experiments.
> Regarding computational complexity of some of the components:
>
> - Optimal‑transport couplings add only linear $O(d)$ overhead. For very high‑dimensional data there is both theoretical and empirical evidence suggesting dropping the solver and using random couplings that do not cost. (please see response **W1** for the reviewer **XTSh**).
>
> - The full Hessian spectrum with $O(d^3)$ is used exclusively for the Local Intrinsic Dimensionality diagnostic and is neither required for training nor for unconditional/controlled generation; practical deployments can drop this step or approximate it as suggested in the limitation section.
>
> We appreciate the reviewer’s comment and will incorporate a deeper exploration of scalability to larger-scale datasets in our outlook and future research directions.
>
> **Q2: The exposition of the results could be clearer. You write that you surpass state-of-the-art EBMs. This is technically true, but somehow omits the fact that other methods in the comparison achieve even better results. I am not in favor of only accepting results that beat state of the art, but you need to argue why your results are still relevant and what advantage/disadvantage other methods have compared to yours.**
>
> We thank the reviewer for pointing out that our comparison to non-EBM methods could indeed be clarified further. The inclusion of non-EBM results primarily serves as a contextual benchmark to illustrate that our method significantly advances energy-based modeling toward competitive generation quality, which was previously challenging for EBMs. Within the category of EBMs, our approach clearly achieves state-of-the-art performance.
> The relevance of EBMs lies in their unique advantages: they inherently provide explicit likelihood estimates (in a single forward pass) for any given sample, enabling straightforward and flexible conditional sampling. Conditions or new interactions between samples can be incorporated by simply adding energies, resulting in a composite energy landscape (explicitly detailed in Algorithm 3 of Appendix B: Posterior Sampling,  Section 3.2 on Inverse Problems, and our response to reviewer **XTSh**'s question **Q2**). Additionally, EBMs naturally provide Hessian-based curvature information about the data manifold, enabling applications (such as LID estimation or more in the response to reviewer **4N5t**(**Q4**)) that flow- or score-based methods cannot directly offer without additional approximations.
>
> However, these advantages come with additional computational overhead during training, which we explicitly acknowledge (see our detailed discussion in **W3** of reviewer **FNAJ**). Whether this overhead is justified depends on the application. For use-cases requiring precise control over generation—such as many scientific applications already outlined in the outlook of our manuscript—this additional cost is typically justified. Similarly, scenarios where conditions, interactions, or labels for conditional generation may not be available at training time, necessitating a model encoding a versatile prior that can accommodate new constraints on demand, further highlight the practical value of our approach.
>
> We sincerely thank the reviewer for emphasizing this important point; we will use the additional page in the camera-ready version to explicitly clarify how to interpret comparisons between EBM and non-EBM methods at the beginning of the results section.

---

> > ### Comment · Reviewer_Rsu1 · 2025-08-02
> > **Thanks for the answer, will raise score**
> >
> > Dear authors, thanks for the clarification. I have raised my score.

---

> ### Author Response · Authors · 2025-08-03
>
> We sincerely thank the reviewer for the feedback and for raising the score. The reviewer's thoughtful comments have helped us improve the manuscript, and we will incorporate the suggested points into the revised version.

---

### Official Review · Reviewer_XTSh · 2025-06-30

**Clarity:** 2
**Significance:** 3
**Originality:** 2
**Rating:** 4
**Confidence:** 4

**Summary:**

This paper introduces "Energy Matching," a generative modeling framework that combines aspects of EBMs and flow-matching approaches. The method proposes learning a single, time-independent scalar potential field that moves samples from a noise distribution to the data distribution. This occurs via a dual-regime process: far from the data manifold, samples follow curl-free, optimal transport-like paths, while near the manifold, an entropic energy term transitions them into a Boltzmann equilibrium, intended to capture the data's likelihood structure. Contributions include reported performance improvements over existing EBMs in generation fidelity on benchmarks like CIFAR-10 and ImageNet. The framework is presented as a prior for solving inverse problems, demonstrated with applications such as controlled inpainting and protein design, including an interaction energy term for diversity. It also enables local intrinsic dimension (LID) estimation by analyzing the learned energy landscape, reportedly using fewer approximations than diffusion models. The authors state these results are achieved without time-conditioning, auxiliary generators, or additional networks.

**Questions:**

Figure 6 in Appendix A.2 illustrates that the choice of the temperature-switching parameter $\tau^* $
  (from Eq. 3) can impact FID scores. Could you elaborate on the sensitivity of the model's final performance and training stability to the precise value of $\tau^* $ and the overall shape/schedule of $\varepsilon(t) $? Is the selection of $\tau^*$ primarily empirical, or are there more principled guidelines for its determination based on dataset characteristics or model behavior?

What is the inference process of this method? Does it need both flow matching part and Langevin MCMC part? If so, would it cost too much time? A standalone, explicit sampling algorithm in the main text would improve the clarity and reproducibility of this paper.

**Ethical Concerns:**

["NO or VERY MINOR ethics concerns only"]

**Final Justification:**

The paper proposed a novel algorithm for EBM. The results show obvious improvement. The theoretical argument about distance concentration in high dimensions is compelling. Yet, it's relatively inefficient in training. So I suggest a borderline accept.

**Limitations:**

Yes.

**Paper Formatting Concerns:**

No formatting concerns.

**Quality:**

2

**Strengths And Weaknesses:**

Strengths
1. The core concept of "Energy Matching" to unify flow matching and Energy-Based Models (EBMs) by learning a single, time-independent scalar potential is a novel contribution.

2. The theoretical development, rooted in the Jordan-Kinderlehrer-Otto (JKO) scheme and first-order optimality conditions, provides a principled foundation for the proposed method. The empirical evaluation is quite comprehensive, covering unconditional image generation (CIFAR-10, ImageNet 32x32), inverse problems (inpainting, protein inverse design), and Local Intrinsic Dimension (LID) estimation.

Weaknesses

1. The paper fails to state how much the modeling relies on the OT solver. The paper does not provide a clear definition of the OT solver. Is the method employed in the paper the unique OT solver? If not, can authors provide ablations on the OT solver?

2. The paper does not provide an analysis of the computational overhead introduced by the OT solver (e.g., what percentage of the training time per iteration is spent on this step). The OT solver itself can be computationally expensive, especially as the batch size increases. This cost is a critical factor for the scalability of the method, especially when training involves the MCMC sampling.

3. The paper forthrightly mentions in its limitations that the method requires extra gradient computation with respect to the input, potentially increasing GPU memory usage (20-40%). This is an important practical consideration for scalability.

---

> ### Author Rebuttal · Authors · 2025-07-30
>
> Thank you for your valuable review and constructive comments, recognizing the strengths of our approach, including its novelty and comprehensive evaluation, and for pointing out areas for improvement regarding the OT solver discussion. Below we address the weaknesses (W) and questions (Q) raised, and we commit to clarifying these points further in the revised version.
>
> **W1: The paper fails to state how much the modeling relies on the OT solver. The paper does not provide a clear definition of the OT solver. Is the method employed in the paper the unique OT solver? If not, can authors provide ablations on the OT solver?**
>
> Remark 2.1 ("OT solver") on page 4 of our manuscript mentions that we use the "POT" linear-programming (LP) backend (Flamary & Courty, JMLR 2021), which computes an exact 1-to-1 OT plan per minibatch (e.g., size 128). We thank the reviewer for pointing out that this explanation might not be sufficiently clear. This choice is not unique; approximations such as Sinkhorn (with thresholding) or random coupling can also be effective.
>
> Let us (over‑idealise) and model a standardised CIFAR‑10 image as a vector $x \in \mathbb{R}^d$ drawn from $\mathcal{N}(0,I_d)$ with  $d = 32 \times 32 \times 3 = 3072$, and pair it with an independent Gaussian‑noise vector $z \sim \mathcal{N}(0,I_d)$.
> Each coordinate of the difference $(z - x)$ is distributed as $\mathcal{N}(0,2)$, thus the squared Euclidean distance satisfies $\|z - x\|\_2^2 \sim 2\chi\_d^2$, with mean $2d$, standard deviation $\sqrt{8d}$, and relative spread $\frac{\sqrt{8d}}{2d} \approx 0.025$. This thin shell means all entries of the cost matrix $C_{ij} = \|z_i - x_j\|_2^2$ are nearly identical, exemplifying the distance‑concentration effect highlighted by [D0]. Consequently, whether we solve the transport exactly (LP), entropically (Sinkhorn), or even use a random matching, the accumulated optimal‑transport cost should be similar. This has been examined empirically:
> - [D1] Table. 5: FID 4.44 (LP) vs 4.46 (random) at 100 steps;
> - our new CIFAR-10 runs: 3.97 → 4.00 at 300 steps.
> - [D1] Fig. D.2: for batch > 16, solver accuracy saturates on the 2‑moons task.
> - [D2] App. C.2: LP and Sinkhorn are indistinguishable unless regularisation is extreme.
>
> We kept POT LP solver for zero hyperparameters, robustness, and negligible cost (see **W2**). We thank the reviewer and will add a dedicated appendix with the discussion.
>
> - [D0] Aggarwal, et al. “On the Surprising Behavior of Distance Metrics in High Dimensional Space” (ICDT 2001)
> - [D1] Tong, et al. “Improving and generalizing flow-based generative models with minibatch optimal transport” (TMLR 2024)
> - [D2] Terpin, et al. “Learning diffusion at lightspeed” (NeurIPS 2024)
>
>
> **W2: The paper does not provide an analysis of the computational overhead introduced by the OT solver (e.g., what percentage of the training time per iteration is spent on this step). The OT solver itself can be computationally expensive, especially as the batch size increases. This cost is a critical factor for the scalability of the method, especially when training involves the MCMC sampling.**
>
> We acknowledge that the original manuscript did not quantify the computational overhead from the OT solver. Additional experiments show that the OT solver accounts for **1.5%** of the training iteration time in Phase 1 (Algorithm 1), dropping to a negligible (**0.01%**) level in Phase 2 (Algorithm 2), where the computational cost is dominated by negative samples generation.
>
> We thank the reviewer and will include this analysis in the revised manuscript.
>
> **W3: The paper forthrightly mentions in its limitations that the method requires extra gradient computation with respect to the input, potentially increasing GPU memory usage. This is an important practical consideration for scalability.**
>
> The extra costs buy a key EBM benefit: $V(x)$ supplies explicit curvature and likelihood information, unlocking inverse‑problem solvers and manifold analysis. Flow/score models lack this and instead resort to heavier inference tricks: full‑trajectory differentiation in [D3] or repeated noise/data alternations in  [D4] which come with their own limitations.
> - [D3] H. Ben-Hamu, et al., “D-Flow: Differentiating through flows for controlled generation,” (ICLR 2024)
> - [D4] M. Mardani, et al., “A variational perspective on solving inverse problems with diffusion models,” (ICLR 2024)
>
>
> **Q1: Figure 6 in Appendix A.2 illustrates that the choice of the temperature-switching parameter  (from Eq. 3) can impact FID scores. Could you elaborate on the sensitivity of the model's final performance and training stability to the precise value of  and the overall shape/schedule of ? Is the selection of primarily empirical, or are there more principled guidelines for its determination based on dataset characteristics or model behavior?**
>
> Thank you for raising this insightful question regarding sensitivity to the temperature-switching parameter ($\tau^* $).
> The parameter $\tau^* $ marks the activation time ($t>\tau^* $) of the Brownian-motion component in our sampling dynamics; prior to this ($t\le \tau^* $), sampling follows deterministic optimal-transport paths toward high-likelihood regions. As described in Algorithm 3 (Appendix B), conditional sampling terms depend explicitly on the temperature $\varepsilon$, vanishing as $\varepsilon \to 0$.
> Lowering $\tau^* $ (e.g., 0.8) allows earlier conditioning, potentially aiding conditional tasks, but typically reduces performance in unconditional generation. Conversely, setting $\tau^* $ closer to 1.0 reinforces OT regularization near the data manifold, stabilizing training and enhancing unconditional generation quality.
> Initially, we presented only a brief ablation of $\tau^* $ (Figure 6, Appendix A.2). Encouraged by your feedback, we've expanded these experiments, individually optimizing hyperparameters (e.g., $\lambda_{cd}$) for different $\tau^* $ values. We now report two hyperparameter sets: Set #1 optimized at $\tau^* =0.9$, and Set #2 optimized specifically at $\tau^* =1.0$.
>
> Our updated CIFAR-10 results are:
> | $\tau^*$ | FID ↓  | Notes                                |
> |----------|--------|--------------------------------------|
> | 0.8      | 4.01   | Plateau at ~275 steps (HP set #1)    |
> | 0.9      | 3.97   | Plateau at ~300 steps (HP set #1)    |
> | 1.0      | 3.98   | Plateau at ~300 steps (HP set #1)    |
> | 1.0      | 3.34   | Plateau at ~325 steps (HP set #2)    |
>
> Based on these expanded experiments, we improved upon our originally submitted results (FID 3.97 $\to$ 3.34); the updated result (FID 3.34) will be included in the camera-ready version.
>
> Our recommended heuristic is to always start with $\tau^* =1.0$ for unconditional generation tasks, ensuring the best balance of training stability and final performance. If additional conditions are present, one may explore lower values ($\tau^* < 1.0$) later, carefully examining if they provide benefits. We will explicitly incorporate this heuristic rationale into the camera-ready version.
>
>
> **Q2: What is the inference process of this method? Does it need both flow matching part and Langevin MCMC part? If so, would it cost too much time? A standalone, explicit sampling algorithm in the main text would improve the clarity and reproducibility of this paper.**
>
> We thank the reviewer for the valuable suggestion; our inference procedure smoothly transitions from deterministic sampling (akin to flow-matching, OT paths and no randomness) to stochastic Langevin MCMC sampling, following a discretized temperature schedule $\epsilon^{(n)}$. Specifically, when the temperature is zero ($\epsilon^{(n)}=0$ for $n \Delta t<\tau^* $), the updates are deterministic (flow-matching), and when the temperature becomes positive ($\epsilon^{(n)}>0$ for $n \Delta t \geq \tau^* $), the updates introduce stochastic noise for modes exploration, becoming Langevin MCMC. This unified scheme is explicitly described in Algorithm 3 (currently in Appendix B.1, which we will move to the main text for clarity).
> Specifically, at each inference step, the updates are:
> - **Unconditional update:**
>
>   $x_m^{(n+1)} \leftarrow x_m^{(n)} - \nabla_x V_\theta(x_m^{(n)})\Delta t + \eta\sqrt{2\epsilon^{(n)}\Delta t}$
>
> - **Conditional update:**
>   The conditional update is identical in form to the unconditional one but uses a composite potential:
>
>   $ V_\theta^{(composit)}(x_m^{(n)}) = V_\theta(x_m^{(n)}) +\frac{\epsilon^{(n)}}{\zeta^2}\|y - A(x_m^{(n)})\|^2 + \frac{\epsilon^{(n)}}{\sigma^2}\sum_{k \neq m} W(x_m^{(n)}, x_k^{(n)})$
>
> Here, briefly:
> - $V_\theta(x)$ is the learned scalar potential ($\sim$ negative log-likelihood).
> - $A(\cdot)$ denotes a forward measurement operator used in inverse problems.
> - $W(\cdot,\cdot)$ represents an optional interaction energy between different samples, e.g. encouraging diversity among generated solutions.
> - $\varepsilon^{(n)}$ controls the stochasticity of sampling ($\varepsilon^{(n)}>0$ corresponds to Langevin MCMC).
> - $\eta\sim\mathcal{N}(0,I)$ introduces Gaussian noise at each Langevin step.
>
> CIFAR-10 unconditional (batch size of 128, NVIDIA R6000 GPU), using Euler-Heun (325 steps) and Euler-Maruyama (1000 steps):
>
> | Method           | Params | Steps | Sampling time [s] ↓ | FID ↓               |
> |------------------|--------|-------|---------------------|---------------------|
> | OT-FM            | 37M    | 325   | **88**              | 4.04                |
> | DDPM++           | 62M    | 1000  | 183                 | 3.74                |
> | OT-FM            | 37M    | 1000  | 136                 | 3.45                |
> | Energy Matching  | 50M    | 325   | 173                 | **3.34**$^\dagger$  |
>
> $^\dagger$ as in **Q1**: latest improved result for our Energy Matching model (to appear in camera-ready, same architecture).
>
> We thank the reviewer for suggesting this improvement and will explicitly include Algorithm 3 in the main text to enhance clarity and reproducibility.

---

> > ### Comment · Reviewer_XTSh · 2025-08-03
> >
> > Thank you for the rebuttal addressing my concerns.
> >
> > I appreciate your commitment to add a dedicated appendix with ablation studies. The theoretical argument about distance concentration in high dimensions is compelling. I look forward to seeing the concrete ablation results you mentioned.
> >
> > Your clarification that sampling only uses Langevin dynamics (not both flow matching and MCMC) resolves my concern about inference complexity. Moving Algorithm 3 to the main text as promised will improve clarity and reproducibility.
> >
> > While you provide concrete timing comparisons, the data actually reveals a concerning trade-off: Energy Matching requires nearly 2x the sampling time compared to OT-FM. This computational overhead, combined with the 35% increase in parameters, suggests that the method's practical applicability may be limited by its sampling efficiency.

---

> ### Author Response · Authors · 2025-08-03
>
> We thank the reviewer again for the thoughtful feedback and are glad our clarifications were helpful. We will incorporate this discussion into the revised manuscript.
>
> Although Energy Matching is slower than OT-FM due to gradient-based velocity parameterization (requiring both forward and backward passes, an inherent ~2x theoretical slowdown for EBMs), the scaling remains favorable as both passes scale linearly with the number of parameters. Moreover, it remains competitive with diffusion models and significantly outperforms SOTA diffusion-EBM hybrids [C1] in both parameter efficiency (50M vs. 150M) and sample quality (FID 3.34 vs. 3.68).
>
> Energy Matching (and any other EBM method) may not be optimal for raw sampling speed, but its computational cost aligns with recent trends in generative AI towards scaling inference to unlock new capabilities [C2, C3].
>
> If further clarification would be helpful, we are happy to discuss.
>
> [C1] Y. Zhu, et al. “Learning Energy-Based Models by Cooperative Diffusion Recovery Likelihood” (NeurIPS 2023)
>
> [C2] B. Zhang, et al. "Improving Diffusion Inverse Problem Solving with Decoupled Noise Annealing" (CVPR 2025 oral)
>
> [C3] Nanye Ma, et al. "Inference-Time Scaling for Diffusion Models beyond Scaling Denoising Steps" (2025)

---

> > ### Author Response · Authors · 2025-08-06
> > **Energy Matching & EBM sampling performance**
> >
> > To re-clarify the remaining concern regarding sampling overhead relative to OT-FM (325 steps):
> >
> >
> > - OT-FM uses a non-likelihood-based velocity parameterization, achieves 20% worse FID, and serves as a reference to confirm that our parametrization (forward + backward pass for computing $\nabla_x$) closely aligns with the theoretical ~2× per step overhead compared to velocity-based methods (forward pass only). This per step overhead is intrinsic to EBMs (already widely recognized for their unique probabilistic inference capabilities) and the ~2× factor does not depend on the parameter count of the network as both passes scale equally.
> >
> >
> > - Our primary competitors are state-of-the-art EBMs [K1] and diffusion-EBM hybrids [K2]. We demonstrate substantial improvements over these methods (FID 3.34 ours vs. 8.61 [K1], 3.68 [K2]), achieving, for the first time, generation quality competitive with diffusion- and flow-based models. Notably, this improvement over [K2] is obtained despite using significantly fewer parameters (50M ours vs. 150M theirs) and a time-independent network.
> >
> >
> > - Our contribution does not add any overhead compared to standard EBMs during sampling.
> >
> >
> > We hope this clarifies the concern; please let us know if any additional clarification is needed. If resolved, we kindly ask you to reconsider the score.
> >
> >
> > [K1] H. Lee, et al. "Guiding Energy-based Models via Contrastive Latent Variables" (ICLR 2023 Spotlight)
> >
> > [K2] Y. Zhu, et al. "Learning Energy-Based Models by Cooperative Diffusion Recovery Likelihood" (NeurIPS 2023 Spotlight)

---

> > > ### Comment · Reviewer_XTSh · 2025-08-08
> > >
> > > Thanks for your rebuttal. I believe the authors have solved my main concerns. I intend to raise my score.

---

> > > > ### Author Response · Authors · 2025-08-08
> > > >
> > > > Thank you for the extended and valuable discussion. As a result of this discussion, in the camera-ready version, we will incorporate: (1) OT solver ablation and extended discussion, (2) Algorithm 3 in the main text (unconditional/conditional sampling), (3) guidelines for selecting the temperature-switching parameter, and (4) additional clarifications regarding sampling efficiency, highlighting trade-offs between direct and likelihood-based velocity parameterizations.

---

### Official Review · Reviewer_4N5t · 2025-07-02

**Clarity:** 3
**Significance:** 3
**Originality:** 3
**Rating:** 4
**Confidence:** 4

**Summary:**

This paper aims to combine the benefits of flow-matching and energy-based models, i.e., they aim to learn a time-independent energy function that determines the data distribution. When noisy samples are far away from the data manifold, the energy descent is more like flow-matching/optimal-transport, displaying straight trajectories; when the samples get close to the data manifold, the sampling process transitions to the Langevin dynamics, which converges to a Boltzmann distribution closer to the data distribution. The method is shown to display advantages in generating images, proteins, and estimating local intrinsic dimensionality.

**Questions:**

* Why is "time-independence" preferred? While it is true that the energy function does not explicitly depend on time, the $\epsilon$ schedule implicitly plays the role of time.
* If the first phase (OT) is accurate, why is the second phase (contrastive objective) even necessary? What is the overhead of the OT solver, as a function of data points? How does it compare to the overhead of distillation (train a first model that builds noise-data pairs, instead of solving the OT problem explicitly)?
* In inference time, the Langevin-type sampling can be slow. How does the performance scale with the number of Langevin steps?
* The Hessian of energy, although expensive to compute, contains valuable information. Using Hessian to estimate the LID is a good example. What other things become possible with energy matching that are impossible/challenging with flow matching?

**Ethical Concerns:**

["NO or VERY MINOR ethics concerns only"]

**Final Justification:**

I would like to thank authors for their clarification. The motivation appears clearer to me. I maintain my positive rating.

**Limitations:**

Yes

**Quality:**

3

**Strengths And Weaknesses:**

Strengths:
* The paper is well-written, pleasant to read
* The idea is novel, interesting, and seems to work

Weaknesses:
* Energy matching needs extra overhead compared to flow matching
* The motivation can be made clearer (see questions)

---

> ### Author Rebuttal · Authors · 2025-07-30
>
> Thank you for the thoughtful review and valuable feedback. Below, we provide point-by-point responses addressing the raised weaknesses (W) and questions (Q). We appreciate your comments on the motivation of certain aspects; we agree that this could be clarified further and commit to improving the paper accordingly in the revised version.
>
> **W1:  Energy matching needs extra overhead compared to flow matching**
>
> We agree with the reviewer that Energy Matching introduces additional overhead compared to flow matching, primarily due to the contrastive-divergence training phase (details in **Q2/Q3**, including complexity and timings). However, this overhead is justified by enabling functionalities beyond unconditional generative tasks. Specifically, the contrastive phase refines the energy landscape to explicitly represent data likelihood, which provides essential advantages including conditional posterior sampling and data manifold curvature information (see detailed motivation in **Q2** and **Q4**). In contrast, flow-based models lose explicit likelihood representation, limiting their applicability in broader probabilistic inference scenarios.
>
> **W2: The motivation can be made clearer**
>
> We appreciate this feedback. To clarify our motivation, we will highlight explicitly in the introduction that while flow- and diffusion-based models effectively generate high-quality samples, they inherently lack explicit modeling of the unconditional data score. This absence complicates controlling the generation, likelihood estimation, and data manifold curvature analysis (more details in **Q4**) —limitations that Energy Matching directly addresses.
> We will further emphasize this motivation clearly in the revision.
>
> **Q1: Why is "time-independence" preferred? While it is true that the energy function does not explicitly depend on time, the schedule implicitly plays the role of time.**
>
> The data density we ultimately care about is time-independent; thus, constraining the energy to be time-independent eliminates unnecessary degrees of freedom. A single set of weights covers all noise scales, leading to fewer parameters and improved regularization—our model uses 50M parameters compared to 150M in [C1], yet achieves better (lower) FID scores: 3.34 (our latest CIFAR-10 result, to appear in the camera-ready, more in **Q3**) versus their 3.68. At inference, this approach also removes the need to precisely estimate the noise level, a challenging issue in inverse problems [C2], since estimating time for the schedule $\varepsilon(t)$ is not critical: it is mostly a piecewise constant schedule and inaccuracies in estimating $\varepsilon(t)$ should not significantly impact sample behavior for $t \gg 1$. Thus, parameterizing the potential energy as time-independent provides a minimal representation from a theoretical perspective and robustness in practice.
>
> - [C1] Y. Zhu, et al. “Learning Energy-Based Models by Cooperative Diffusion Recovery Likelihood” (NeurIPS 2023)
> - [C2] H. Chung, et al.  “Diffusion Posterior Sampling for General Noisy Inverse Problems” (ICLR 2023)
>
> **Q2: If the first phase (OT) is accurate, why is the second phase (contrastive objective) even necessary? What is the overhead of the OT solver, as a function of data points? How does it compare to the overhead of distillation?**
>
> We thank the reviewer for this important question. Indeed, as shown in Figure 5a, the OT phase alone effectively aligns transport paths, efficiently guiding samples from noise toward the data distribution. However, the resulting potential from this first phase is not directly meaningful in terms of data likelihood—it merely represents a low-curvature landscape that facilitates efficient transport. Crucially, this learned potential after the OT phase does not encode likelihood of the data in any meaningful way.
> The second phase (CD objective) is therefore necessary to explicitly encode likelihood information into the energy landscape. This is achieved by exploring the data manifold from multiple directions via negative sampling, creating distinct "energy wells" around regions of high data density (Figure 5b). A meaningful potential, proportional to negative log-likelihoods, is the central advantage of energy-based models over flow-based models. This explicit likelihood encoding enables:
> - Flexible conditional generation (e.g. inverse problems, interactions), through posterior sampling.
> - Explicit (in a single forward pass) log-likelihood estimation.
> - Analysis of the data manifold structure (e.g., intrinsic dimensionality estimation).
>
> We agree this motivation could be emphasized more clearly and will explicitly add this discussion to the revised manuscript.
>
> Regarding the overhead of the OT solver as a function of data points and its comparison to distillation:
> the OT solver we use (POT linear program) has complexity $O(B^2 + d)$, where $B$ is the minibatch size and $d$ is the data dimensionality. In practice, the scaling is mainly driven by the $B$, as $d$ only contributes linearly through the computation of the cost matrix, which is relatively inexpensive. For batches well beyond the size of 128 used here, regularized Sinkhorn methods can reduce complexity to $O(B\log B + d)$. Ablation studies in existing literature [C3, Appendix C.2, Figure 10] indicate minimal differences between these OT methods, provided the regularization does not severely degrade the transport map.
> The OT solver accounts for **1.5%** of the training iteration time in phase 1 (Algorithm 1). This drops to **0.01%** during the main training phase (Algorithm 2), after the contrastive-divergence term, which is significantly more expensive due to the simulation-based generation of negative samples, is activated.
> Training a dedicated distilled model to approximate the global OT map presents a promising direction for achieving sampling with fewer steps. Indeed, precomputing the OT map upfront could eliminate sampling errors that arise when the transport map is supervised only on minibatches. We agree this warrants a deeper discussion and promise to add an additional remark on this topic in the revised version.
>
> **Q3: In inference time, the Langevin-type sampling can be slow. How does the performance scale with the number of Langevin steps?**
>
> Please refer to Figure 6 in our manuscript, which shows the relationship between FID and the number of Langevin steps (fixed step size $dt = 0.01$). We observe rapid improvement initially, with performance saturating after about 300–325 steps. While Langevin dynamics: $x_m^{(n+1)} \leftarrow x_m^{(n)} -  \nabla_x V_\theta(x_m^{(n)}) \Delta t + \eta\sqrt{2\varepsilon^{(n)}\Delta t},\quad\eta\sim\mathcal{N}(0,I)$ can require many steps to reach equilibrium, our training ensures rapid convergence near the data manifold (via the OT flow objective), after which Brownian motion efficiently explores data modes.
>
> **Concrete timing results:** Unconditional CIFAR-10 sampling (batch size 128, NVIDIA R6000 48GB GPU), using Euler-Heun (325 steps) and Euler-Maruyama (1000 steps):
>
> | Method           | Params | Steps | Sampling time [s] ↓ | FID ↓               |
> |------------------|--------|-------|---------------------|---------------------|
> | OT-FM            | 37M    | 325   | **88**              | 4.04                |
> | DDPM++           | 62M    | 1000  | 183                 | 3.74                |
> | OT-FM            | 37M    | 1000  | 136                 | 3.45                |
> | Energy Matching  | 50M    | 325   | 173                 | **3.34**$^\dagger$  |
>
> $^\dagger$ FID 3.34 (lower is better) is our latest CIFAR-10 result using energy matching. The main improvements from the originally submitted result (FID 3.97) are due to adjusted hyperparameters: $\tau$* increased to 1.0, $\lambda_{cd} = 1×10^\{-3}$, and the Langevin steps increased from 300 to 325.
>
> We thank the reviewer for the important suggestion. We'll highlight our method's competitive sampling efficiency compared to SOTA flow- and score-based baselines in the revision.
>
> **Q4: The Hessian of energy, although expensive to compute, contains valuable information. Using Hessian to estimate the LID is a good example. What other things become possible with energy matching that are impossible/challenging with flow matching?**
>
> We appreciate this insightful question highlighting a key distinction: energy-based methods explicitly model a scalar field representing data likelihood, unlike flow-based methods. This explicit potential enables probabilistic reasoning tasks (e.g., intrinsic dimensionality estimation, inverse problems, out-of-distribution detection, model selection), which flow-matching inherently lacks. Energy Matching addresses this by learning a potential proportional to negative log-likelihood, supporting likelihood-based computations.
> Explicit energy modeling enables various Hessian-based probabilistic tasks, e.g.:
> - Active-subspace dimension inference [C3]: accelerating inference by identifying data-relevant directions.
> - Laplace approximations/Bayes factors [C4]: efficient model selection via marginal likelihood estimation.
> - Stochastic-Newton preconditioned MCMC [C5]: accelerated sampling, especially useful in high-dimensional inverse problems.
>
> We thank the reviewer for highlighting this important consideration. While we acknowledge Energy Matching’s higher training cost due to the CD phase (as further elaborated in our response to **W3** of reviewer **FNAJ**), we agree that clearly outlining this trade-off is important. We will emphasize the probabilistic advantages, particularly in likelihood-based reasoning tasks, more explicitly in the revised manuscript.
>
> - [C3] P. Constantine, et  al.  “Accelerating Markov Chain Monte Carlo with Active Subspaces” (SIAM 2015)
> - [C4] E. Daxberger, et al. "Laplace redux-effortless Bayesian deep learning." (NeurIPS 2021)
> - [C5] T. Bui-Thanh, et al. “A computational framework for infinite-dimensional Bayesian inverse problems” (SIAM 2015)

---

> > ### Author Response · Authors · 2025-08-05
> >
> > Dear Reviewer 4N5t,
> >
> > Thank you once again for your thoughtful feedback and insightful questions. In our rebuttal, we clarified the motivation for using a time-independent energy function, detailed the overhead introduced by the contrastive-divergence phase, showcased competitive sampling efficiency compared to baseline methods, and elaborated on the probabilistic benefits enabled by energy matching. We hope these explanations addressed your concerns clearly. If there are any points that would benefit from further clarification or additional discussion, please let us know. We greatly appreciate your input.
> >
> > Thank you,
> > The Authors

---

> > > ### Comment · Reviewer_4N5t · 2025-08-08
> > >
> > > I would like to thank authors for their clarification. The motivation appears clearer to me. I maintain my positive rating.

---

> > > > ### Author Response · Authors · 2025-08-09
> > > >
> > > > Thank you for your thoughtful feedback, which has helped us improve the manuscript. We will clarify the motivation for the time-independent potential and highlight how the contrastive phase enables additional probabilistic reasoning tasks in the revised manuscript.

---

### Note · Authors · 2025-08-11

We sincerely thank the reviewers for their valuable feedback, significantly improving our manuscript. All reviewers recognized the novelty and significance of Energy Matching, describing it as “novel, interesting work” (4N5t), a “relevant contribution” (Rsu1), having a “principled foundation” (XTSh), and effectively “integrating optimal transport-based flows with EBMs” (FNAJ). The rebuttal primarily focused on implementation details, efficiency, and scalability.

**Before the rebuttal:**

Reviewers requested clarifications on our likelihood-based velocity parametrization, training and inference efficiency, scalability, OT solver choice, and heuristic selection of the phase-switching hyperparameter.

**During the rebuttal:**

We clarified motivations behind our likelihood-based velocity parametrization, emphasizing its importance for precise posterior sampling and direct manifold curvature learning, tasks fundamentally challenging for flow or diffusion-based methods. We explicitly addressed efficiency, highlighting that our gradient-based parametrization scales comparably to direct methods. We provided OT solver ablations, noting minimal complexity mainly relevant in lower dimensions, and offered heuristic guidelines for phase-switching. Inference details previously in the appendix were moved to the main manuscript, demonstrating seamless transition from deterministic optimal transport to stochastic MCMC-based exploration.

**Summary:**

Energy Matching learns a potential energy landscape explicitly encoding likelihood information near data, enabling precise generation, while far from data it has minimal curvature, allowing efficient and high-quality sampling without taking detours.  Our approach significantly advances likelihood-based generative modeling, substantially outperforming state-of-the-art EBMs (FID $\downarrow$: 3.34 ours vs. 8.61 [K1]) and diffusion-EBM hybrids (FID$\downarrow$: 3.34 ours vs. 3.68 [K2]), using fewer parameters (50M vs. 150M) in a time-independent network without auxiliary generators. We demonstrated explicit control in inpainting and inverse protein design via a novel inference-time interaction energy, boosting diversity and posterior exploration. Core components scale efficiently in network and data dimensions.

Reviewers welcomed our clarifications, maintaining or raising their assessments.
Energy Matching aligns naturally with recent generative AI trends toward scaling inference for new capabilities [C2, C3] (2025).

---

### Decision · Program_Chairs · 2025-09-17

**Decision:**

Accept (poster)

**Comment:**

Tl;dr: Based on the reviews, rebuttal and ensuing discussion I recommend accept.

### Paper Summary

A novel generative modeling framework, Energy Matching, is introduced, that unifies flow matching and energy based models. Proposed method learns a single time-independent scalar potential energy function that governs the generative process. Paper claims that this approach allows for efficient, transport-like sampling far from the data manifold and explicit likelihood modeling near the data manifold. The empirical results show that Energy Matching significantly outperforms previous EBMs on standard image generation benchmarks (CIFAR-10, ImageNet 32x32) and is competitive with state-of-the-art diffusion and flow-matching models, while offering the additional benefits of a learned likelihood.

### Key strengths and weaknesses

Strenghts: 1) Novelty. Proposed method is novel and principled. The formulation is grounded in the JKO scheme, providing a solid theoretical foundation, 2) Strong empirical results: substantial improvements over existing EBMs are shown with competitive performance against other SOTA methods. Additional results on controlled inpainting and protein design further demonstrate the strengths of the approach.

Weaknesses: 1) Writing:  The main initial weakness (also pointed out by the reviewers) was the clarity of the presentation, particularly for people not familiar with optimal transport theory, 2) Overhead: The computational overhead of the proposed method, especially during training, compared to pure flow-matching models, was another concern, 3) Scaling to higher resolution: The scaling of the method to higher-resolution images was another concern.

### Decision justification

The decision to recommend acceptance is based on the novelty, theoretical elegance, and strong empirical results of the proposed method. The reviewers had concerns regarding the clarity of the method, the computational overhead, the role of the OT solver, and the scalability of the approach. The authors provided detailed and convincing responses to all the concerns. Authors have committed to significant revisions to the paper to improve its clarity, including adding a more intuitive explanation of the method, a more detailed discussion of the computational overhead, and moving the sampling algorithm to the main text. The reviewers were satisfied with the author responses and the planned revisions, and were unanimously in favor of acceptance.